# LDARNet: DNA Adaptive Representation Network with Learnable Tokenization for Genomic Modeling

**Daria Ledneva** [1]   **Denis Kuznetsov** [1]

## Abstract

Genomic foundation models increasingly adopt large language model architectures, yet almost universally rely on fixed tokenization schemes such as $k$-mers, BPE, or single nucleotides, which impose arbitrary sequence boundaries that may obscure biologically relevant structure. We present LDARNet, a 120M-parameter hierarchical genomic foundation model that adapts H-Net-style dynamic chunking from autoregressive generation to masked language modeling, combining BiMamba-2 state-space layers with local attention, bidirectional routing, and a ratio-based regularizer to induce adaptive token boundaries without supervision. Fine-tuned on 27 tasks from the Nucleotide Transformer and Genomic Benchmarks suites, LDARNet achieves 11/18 wins among compact models ($<$300M parameters) and state-of-the-art results on 5 histone modification tasks, outperforming models up to $20\times$ larger. A FLOPs-matched controlled experiment isolates learned routing as the source of these gains: learned boundaries beat fixed-grid boundaries by up to 14 percentage points on histone tasks at identical compute. Nucleotide-resolution analysis further shows that the learned boundaries align with canonical promoter motifs and splice junctions without supervision, providing a biological interpretation for adaptive tokenization in genomic foundation models.

## 1. Introduction

The success of large language models (LLMs) has motivated the development of foundation models for genomics, where large-scale pretraining can transfer across diverse predictive tasks. Recent models such as DNABERT and Nucleotide Transformer (Ji et al., 2021; Lopez et al., 2023) demonstrate that pretrained encoders can generalize effectively to promoters, enhancers, and splice sites. However, most approaches rely on *fixed tokenization*, such as $k$-mers or byte-pair encoding (BPE). While these schemes are effective for text, they impose arbitrary boundaries on genomic data and lack clear biological grounding, raising the question of whether adaptive tokenization can capture functional signals more faithfully.

A recent line of work has begun to address this. H-Net (Hwang et al., 2025) introduced dynamic hierarchical tokenization in an autoregressive (AR) setup, demonstrating that adaptive segmentation is feasible at scale on DNA via reduced perplexity. However, H-Net was not evaluated on downstream classification benchmarks, leaving open whether adaptive tokenization yields embeddings competitive with established genomic foundation models under the masked language modeling (MLM) paradigm.

We address this gap with **LDARNet** (Learnable DNA Adaptive Representation Network), a hierarchical model that adapts the H-Net architecture to MLM pretraining. Our main contributions are:

- We adapt the H-Net dynamic tokenization architecture (Hwang et al., 2025) from autoregressive generation to masked language modeling. Code and model weights will be made available at https://github.com/darlednik/ICML-LDARNet.

- We demonstrate that hierarchical compression with dynamic boundary prediction enables a compact 120M-parameter model to match or surpass baselines 4–20$\times$ larger, achieving best overall performance on 5 histone modification tasks against 2.5B-parameter competitors through comprehensive fine-tuning evaluation across 27 diverse genomic tasks.

- We establish that architectural generality through learnable multi-scale compression preserves cross-species transferability where domain-specific optimization sacrifices it: on the Nucleotide Transformer (NT) benchmark suite (Dalla-Torre et al., 2025), LDAR-

---

[1]Moscow Independent Research Institute of Artificial Intelligence, Moscow, Russia. Correspondence to: Daria Ledneva <a.ledn2026@gmail.com>.

*Proceedings of the $43^{rd}$ International Conference on Machine Learning*, Seoul, South Korea. PMLR 306, 2026. Copyright 2026 by the author(s).

Net achieves 11/18 wins, compared to 1–2 for human-genome-specialized alternatives of comparable scale, while remaining competitive on the human-centric Genomic Benchmarks (GB) suite (Grešová et al., 2023).

- We provide direct, causal evidence that *learned* routing – not the hierarchical scaffolding alone – drives the gains: at matched FLOPs, learned boundaries beat fixed-grid boundaries by up to 14.3pp on histone tasks, and a nucleotide-resolution analysis shows that they land on canonical promoter motifs and splice junctions without supervision.

**Conflict of Interest Disclosure.** The authors declare no financial conflicts of interest. LDARNet is evaluated exclusively on publicly available benchmarks (Nucleotide Transformer tasks, Genomic Benchmarks) maintained by independent research groups, and none of the baseline models compared in this work are developed by the authors' employer.

## 2. Related Works

### 2.1. Technical Foundations

A central challenge for foundation models in genomics and other non-linguistic domains lies in tokenization. Transformers (Vaswani et al., 2017) achieved remarkable success in NLP by operating over subword vocabularies, but this design presumes the existence of semantically meaningful and human-interpretable units such as words. For DNA and raw byte sequences, where no such segmentation exists, tokenization remains an open problem: fixed schemes such as $k$-mers introduce arbitrary boundaries, while byte-level encodings dramatically inflate sequence length.

Several works attempted to bypass tokenization through isotropic byte-level modeling. MambaByte (Wang et al., 2024) applied Mamba-2 layers directly to characters, while LlamaByte extended Transformers to raw sequences. Although these approaches eliminate external preprocessing, flat byte-level models typically underperform tokenized counterparts of comparable scale, suggesting that meaningful intermediate units are still needed. SpaceByte (Slagle, 2024) partially addressed this by introducing hand-crafted boundary heuristics (e.g., space delimiters) to form chunks, but such strategies remain domain-specific and inflexible.

The H-Net framework (Hwang et al., 2025) reframed tokenization as a learnable problem, introducing dynamic chunking that jointly optimizes boundary detection and representation learning in a multi-stage hierarchy. By replacing the traditional tokenization–LM–detokenization pipeline with an end-to-end architecture, H-Net demonstrated that adaptive chunking can outperform tokenized Transformers at comparable scale and improve data efficiency in settings with weak or arbitrary tokenization heuristics. These developments highlight tokenization not as a preprocessing choice but as a central modeling challenge, motivating architectures that can learn biologically meaningful units directly from raw sequences.

### 2.2. DNA Foundation Models

Large-scale self-supervised pretraining has been rapidly adopted in genomics, giving rise to a family of DNA foundation models. Early contributions such as DNABERT (Ji et al., 2021) demonstrated the utility of BERT-style masked language modeling on genomic data using fixed $k$-mer tokenization, establishing a strong baseline for sequence-based prediction tasks. Subsequent works such as the Nucleotide Transformer (NT) (Lopez et al., 2023) and its successor NTv2 (Dalla-Torre et al., 2025) scaled Transformer encoders from hundreds of millions to billions of parameters trained across multi-species genomes, demonstrating strong transferability of genomic embeddings but facing the quadratic context-length bottleneck inherent to attention. To mitigate this limitation, several works integrated more efficient sequence architectures. GENA-LM (Fishman et al., 2025) employed sparse attention to extend receptive fields, while Caduceus (Schiff et al., 2024) introduced BiMamba blocks with shared weights, leveraging state-space recurrence for efficient long-context modeling. HyenaDNA (Nguyen et al., 2023) proposed implicit long convolutions that support substantially longer contexts, and JanusDNA (Duan et al., 2025) combined AR efficiency with the bidirectionality of masked modeling in a hybrid Mamba–Attention Mixture-of-Experts design, enabling pretraining on million-base sequences. Together, these architectures illustrate the trade-off between capacity, efficiency, and context length that continues to shape genomic foundation model design.

### 2.3. Tokenization in Genomic Models

Tokenization remains a central open problem in genomic modeling, as DNA sequences lack the explicit segmentation cues present in natural language. Fixed $k$-mer approaches (Ji et al., 2021; Lopez et al., 2023) established strong early baselines but impose boundaries that are not explicitly biologically informed. Byte-level models (Nguyen et al., 2023; Schiff et al., 2024; Duan et al., 2025) preserve nucleotide-level resolution, but increase the effective sequence length and computational cost, while leaving higher-order compositional units to be inferred implicitly by the model. Subword strategies based on BPE (Zhou et al., 2023; Fishman et al., 2025) introduce variable-length units, but their vocabularies are induced from statistical co-occurrence patterns and are not explicitly constrained to correspond to biological units.

Recent work has begun to explore adaptive tokenization for genomic sequences. MxDNA (Qiao et al., 2024) learns

discontinuous and overlapping units through a mixture-of-experts convolutional design, while VQDNA (Li et al., 2024) uses vector quantization to induce hierarchical genomic vocabularies. Neither model is included in our downstream baselines: for MxDNA, adapting the released architecture to the Generator (Wu et al., 2025) fine-tuning protocol required unspecified implementation choices, while VQDNA did not provide a complete implementation for a matched evaluation at the time of writing. These methods show that learned tokenization can improve downstream performance and capture biologically relevant patterns. However, they do not formulate tokenization as an explicit boundary-placement problem: MxDNA analyzes mixture-of-experts assignments and token embeddings, while VQDNA analyzes codebook structure, but neither validates boundary locations against canonical genomic landmarks. LDARNet addresses this gap by adapting H-Net-style hierarchical chunking to masked language modeling and making learned boundaries themselves an object of biological analysis.

# 3. LDARNet Architecture

We introduce LDARNet, a hierarchical foundation model for genomic sequences that extends the H-Net design (Hwang et al., 2025) with several architectural innovations. While H-Net was originally developed for AR language modeling, our modifications adapt the framework to MLM and introduce bidirectional mechanisms that better align with the bidirectional nature of DNA.

At a high level, LDARNet retains the hierarchical encoder–main–decoder organization of H-Net but incorporates four major changes: (i) Mamba layers are replaced with *BiMamba-2* blocks with shared weights, extending the Bi-Mamba design of Caduceus (Schiff et al., 2024) to Mamba-2; (ii) all attention mechanisms are non-causal; (iii) the encoder additionally includes a single local attention layer for fine-grained pattern recognition, while the backbone and decoder are instantiated entirely with BiMamba-2 blocks; and (iv) both the router and dechunking modules are extended to bidirectional variants. This design preserves H-Net's efficiency while improving expressivity and stability for genomic modeling.

## 3.1. Overview

Like H-Net, LDARNet consists of stacked stages of encoders, a central backbone, and decoders, as Figure 1 illustrates. Each stage compresses the sequence through a learned, content-aware *chunking* operation (Section 3.3), processes it at a reduced resolution, and restores fine-grained information through *dechunking*.

For an $S$-stage hierarchy, we denote encoders and decoders by $\mathcal{E}^s$ and $\mathcal{D}^s$ ($1 \leq s \leq S$), and the central backbone by $\mathcal{M}$.

The overall forward process at each stage $s$ is:

$$\hat{x}^{s+1} = \mathcal{E}^s(x^s), \quad x^{s+1} = \mathsf{Chunk}(\hat{x}^{s+1}), \quad (1)$$

followed by backbone processing $\hat{z}^S = \mathcal{M}(x^S)$ at the innermost stage, and reconstruction

$$z^s = \mathsf{Dechunk}(\hat{z}^s), \quad \hat{z}^{s-1} = \mathcal{D}^s(z^s), \quad (2)$$

on the way back up. The Chunk and Dechunk modules are defined in Section 3.3. Unlike H-Net, all layers in LDARNet use bidirectional (non-causal) context, as required by the MLM objective.

## 3.2. Sequence Processing Blocks

### 3.2.1. ENCODER AND DECODER BLOCKS: BIMAMBA-2

We replace the causal Mamba layers in H-Net's outer networks with a *bidirectional*, non-causal variant of Mamba-2, which we term **BiMamba-2**, extending the BiMamba design of Caduceus (Schiff et al., 2024) to Mamba-2. This design preserves the linear-time recurrence of state-space models while enabling full-context conditioning, which is essential for MLM and other non-autoregressive objectives.

**Mamba-2 as selective state-space layers.** Mamba-2 (Dao & Gu, 2024) instantiates a *selective* state-space layer whose dynamics are conditioned on the input. The model admits both a linear recurrent formulation and a quadratic dual representation via structured semiseparable (SS) matrices, a property referred to as SSD duality. For input $x_t \in \mathbb{R}^D$ and hidden state $h_t \in \mathbb{R}^N$:

$$h_{t+1} = \bar{A}_t h_t + \bar{B}_t x_t, \quad y_t = C_t h_t + D x_t, \quad (3)$$

$$\bar{A}_t, \ \bar{B}_t = \mathrm{discretize}(A, \ B_t, \ \Delta_t), \quad (4)$$

$$B_t = W_B x_t, \ \ C_t = W_C x_t, \ \ \Delta_t = \mathrm{softplus}(W_\Delta x_t) \quad (5)$$

Efficient GPU kernels implement block-SS decompositions and fused projections, supporting large $N$ with stable training and favorable wall-clock efficiency.

**Bidirectional construction with mean fusion.** To enable bidirectional context aggregation, we apply a Mamba-2 (M2) cell in both the forward and reversed temporal order, fusing the outputs by mean-pooling. We adopt mean rather than sum fusion based on our ablation (Appendix B.3), which shows mean fusion gives stronger performance on H3K4me1, H3K4me3, and H3K36me3 – precisely the tasks on which LDARNet achieves its largest gains relative to larger baselines. Given input $X \in \mathbb{R}^{B \times T \times D}$ and padding mask $M \in \{0, 1\}^{B \times T}$:

$$Y = \tfrac{1}{2}\Big[ \mathrm{M2}(X \odot M) + \mathrm{flip}_T\big(\mathrm{M2}(\mathrm{flip}_T(X \odot M))\big) \Big], \quad (6)$$

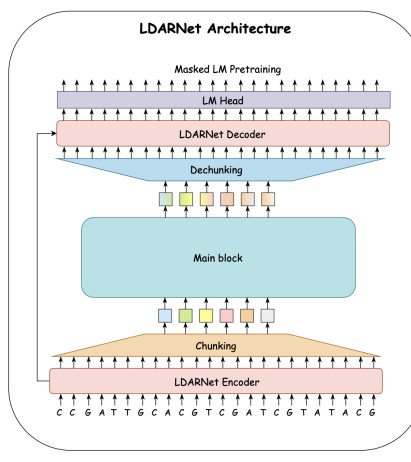 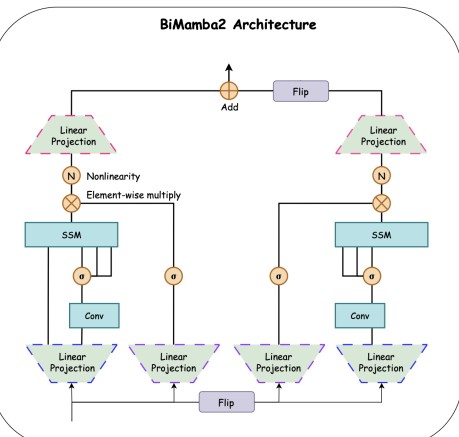

*Figure 1.* **Model overview.** Left: the LDARNet architecture with BiMamba-2 outer layers and a BiMamba-2 backbone operating in a compressed latent space; the encoder additionally includes one local attention layer for fine-grained pattern recognition. Right: the internal structure of a BiMamba-2 block used throughout the network.

where $\odot$ applies masking across features (broadcasting over $D$) and $\text{flip}_T$ denotes temporal reversal. Mean fusion avoids introducing additional parameters while preserving symmetry across directions.

**Parameter tying.** A naive bidirectional construction would double the number of parameters, since separate forward and reverse Mamba-2 modules would each maintain their own projections. However, most parameters reside in the input and output projection layers rather than in the convolution or SSM submodules (Gu & Dao, 2024). To avoid this overhead, BiMamba-2 shares these dominant weights across directions:

$$\{W_{\text{in}}, b_{\text{in}}, W_{\text{out}}, b_{\text{out}}\}^{\rightarrow} \ = \ \{W_{\text{in}}, b_{\text{in}}, W_{\text{out}}, b_{\text{out}}\}^{\leftarrow}. \quad (7)$$

This yields a parameter-efficient block in which the forward and reverse passes instantiate a single shared Mamba-2 definition. We note that this provides only directional symmetry: it does not architecturally encode reverse-complement (RC) equivalence (i.e., the biological symmetry A↔T, C↔G between a strand and its complement). To encourage RC consistency, we additionally apply reverse-complement augmentation during pretraining (Section 3.5).

### 3.2.2. MAIN BACKBONE.

The core backbone $\mathcal{M}$ operates on compressed representations ($L^S \ll L^0$), where each latent token aggregates information from a variable-length segment of the original sequence. We instantiate $\mathcal{M}$ with BiMamba-2 layers, replacing the Transformer backbone used in the original H-Net formulation.

This choice is motivated by the architectural ablation in Appendix B.1. Under the notation introduced there, our

selected configuration `mt-M-m` achieves the best average performance across the four evaluated histone modification tasks, and is individually best on H3K4me1 and H3K36me3. In contrast, the Transformer-backbone configuration `mt-T-m` performs substantially better on splice-site classification (0.964 vs. 0.901 MCC), with a modest reduction on histone tasks. These results indicate a task-dependent trade-off: Transformer backbones better preserve fine-grained positional information for splice-site prediction, whereas the BiMamba-2 backbone is more favorable for long-range epigenetic modeling. We adopt the BiMamba-2 backbone in the main LDARNet configuration, consistent with the model's strongest empirical regime on histone modification tasks.

The encoder additionally includes a single local attention layer, which provides a complementary fine-scale inductive bias for short-range motif recognition; its contribution is evaluated in Appendix B.1.

### 3.3. Dynamic Chunking and Dechunking

A central innovation of our architecture lies in adapting the dynamic chunking mechanism of H-Net (Hwang et al., 2025) to a *bidirectional MLM setting*. We introduce two key modifications: (i) the *router* employs bidirectional similarity to detect boundaries symmetrically and robustly to masked input positions, and (ii) the *dechunker* incorporates a bidirectional exponential moving average (EMA) smoother that propagates information into masked positions from both directions.

**Bidirectional Routing.** Given encoder outputs $\hat{x}_t \in \mathbb{R}^D$, the router projects them into query–key pairs,

$$q_t = W_q \hat{x}_t, \quad k_t = W_k \hat{x}_t, \quad (8)$$

where $W_q, W_k \in \mathbb{R}^{D \times D}$ are initialized as identity matrices, so that early similarity scores reflect the encoder representation geometry and stabilize boundary statistics.

Unlike the unidirectional routing rule in H-Net, we estimate boundary evidence using both orientations of each adjacent pair. For the candidate boundary between positions $t-1$ and $t$, we compute (for $t > 1$):

$$s_t^{\text{fwd}} = \cos(q_{t-1}, k_t), \quad s_t^{\text{bwd}} = \cos(q_t, k_{t-1}), \quad (9)$$

$$\bar{s}_t = \tfrac{1}{2}(s_t^{\text{fwd}} + s_t^{\text{bwd}}), \quad p_t = \tfrac{1}{2}(1 - \bar{s}_t). \quad (10)$$

Since cosine similarity lies in $[-1, 1]$, this gives $p_t \in [0, 1]$, with low similarity between adjacent representations yielding a stronger boundary signal. Position $t = 1$ is treated as a boundary by construction ($p_1 := 1$), so that each sequence has a well-defined initial chunk. From these probabilities, we derive a hard boundary mask

$$b_t = \mathbb{1}_{\{p_t > 0.5\}}, \quad (11)$$

following the threshold convention of H-Net (Hwang et al., 2025). Invalid padded positions are excluded from the mask, preventing chunks from crossing padded regions.

This bidirectional formulation is particularly important under MLM: since approximately $15\%$ of input positions are replaced by [MASK], a unidirectional router can receive a corrupted similarity signal near candidate boundaries. Averaging over both orientations of each adjacent pair allows boundary decisions to rely on context from both sides, making routing less sensitive to local masking artifacts.

**Chunking.** The chunking operator compresses the sequence by retaining the representations at positions selected by the boundary mask, while preserving their relative order:

$$x^{s+1} = \big(\hat{x}_t\big)_{t:b_t=1}, \quad p^{s+1} = \big(p_t\big)_{t:b_t=1}. \quad (12)$$

This defines a data-dependent hierarchical downsampling mechanism: compressed tokens are placed at learned boundary positions rather than at fixed strides.

**Bidirectional Dechunking with EMA.** After the compressed sequence has been processed by the backbone, its representations must be expanded back to the resolution of the previous hierarchy level. This reconstruction is sensitive to discrete boundary decisions: each compressed token summarizes a variable-length span, and errors in boundary placement can affect all positions assigned to that span. To make this expansion more stable, we extend the H-Net dechunker with a bidirectional EMA-style smoother.

Let $z_j$ denote the backbone output at compressed position $j$, and let $p_j$ be the corresponding boundary probability retained during chunking. We propagate information in both directions using EMA-like dynamics and average the two passes:

$$\bar{z}_j^{\text{fwd}} = p_j z_j + (1 - p_j)\bar{z}_{j-1}^{\text{fwd}},$$
$$\bar{z}_j^{\text{bwd}} = p_j z_j + (1 - p_j)\bar{z}_{j+1}^{\text{bwd}}, \quad (13)$$
$$\bar{z}_j = \tfrac{1}{2}(\bar{z}_j^{\text{fwd}} + \bar{z}_j^{\text{bwd}}).$$

This recurrence is implemented efficiently via a fused selective-scan kernel parameterized by the boundary probabilities.

Bidirectional propagation is particularly important under MLM: masked positions may lie inside a learned chunk, where unidirectional dechunking would propagate information from only one side. The two-sided pass allows reconstructed representations to incorporate context from compressed states on both sides, making the expansion less sensitive to local masking artifacts.

**Upsampling.** To restore the sequence to its previous length $L^s$, each original position is assigned the smoothed representation of the most recent selected boundary token:

$$\tilde{z}_t = \bar{z}_{j(t)}, \qquad j(t) = \sum_{k=1}^{t} b_k, \quad (14)$$

where $j(t)$ is the cumulative number of selected boundaries up to position $t$. Each compressed token is thus broadcast over the span between its boundary and the next, giving a piecewise-constant expansion back to the original resolution.

### 3.4. Training Objective

Unlike H-Net's AR training, LDARNet is optimized under an MLM loss, which is more appropriate for bidirectional DNA modeling: regulatory elements such as transcription factor binding sites and splice junctions are read by cellular machinery from either direction, motivating an objective that exposes the model to bidirectional context at training time. The overall objective is:

$$\mathcal{L} = \mathcal{L}_{\text{MLM}} + \alpha \sum_{s=0}^{S-1} \mathcal{L}_{\text{ratio}}^s, \quad (15)$$

where the first term is the standard cross-entropy loss for MLM, and the second term regularizes the compression ratio at each stage to avoid degenerate chunking solutions.

**Ratio Loss.** We adopt the ratio loss from H-Net (Hwang et al., 2025), originally introduced to prevent trivial compression behavior:

$$\mathcal{L}_{\text{ratio}} = \frac{N}{N-1}\big((N-1)FG + (1-F)(1-G)\big), \quad (16)$$

$$F = \frac{1}{L} \sum_{t=1}^{L} b_t, \quad G = \frac{1}{L} \sum_{t=1}^{L} p_t, \qquad (17)$$

where $F$ is the fraction of vectors actually selected, $G$ is the average boundary probability over the sequence of length $L$, and $N$ is the target compression ratio. By construction, the minimum $\mathcal{L}_{\text{ratio}} = 1$ is attained at $F = G = 1/N$. Without this regularizer, the router has no intrinsic tendency to match the target compression: at $N = 4$ with $\alpha = 0$, the effective compression drifts throughout training while the MLM loss remains essentially unchanged from the regularized run (Appendix B.4). The ratio loss thus stabilizes compression at no cost to reconstruction quality.

### 3.5. Model Implementation and Pretraining

We instantiate **LDARNet** as a 120M-parameter single-stage hierarchical model with compression ratio $N = 4$. The architecture follows an encoder–backbone–decoder structure `[m3t1, [M10], m4]`: three BiMamba-2 layers and one local attention layer in the encoder, ten BiMamba-2 layers in the backbone, and four BiMamba-2 layers in the decoder. Model dimensions follow the H-Net convention of widening compressed stages (Hwang et al., 2025): $d_{\text{model}} = 512$ in outer stages and $d_{\text{model}} = 768$ in the backbone. The vocabulary comprises seven byte-level tokens: {A, C, G, T, N, `[PAD]`, `[MASK]`}.

**Training.** We employ MLM with 15% masking probability, combining the reconstruction loss with the ratio-based regularizer ($\alpha = 0.03$) introduced in Section 3.4. Models are optimized using AdamW (Loshchilov & Hutter, 2017) with base learning rate $5 \times 10^{-4}$ and a warmup-stable-decay (WSD) schedule: 10% warmup, 70% plateau, 20% decay. Following Hwang et al. (2025), we apply stage-wise learning rate scaling to outer layers ($3\times$ multiplier) to compensate for gradient attenuation through the compressed backbone. Training uses effective batch size 512 on sequences of length 4096.

**Corpus.** The pretraining data combines the human reference genome with the multispecies collection from Nucleotide Transformer (Dalla-Torre et al., 2025), ensuring both in-species fidelity and cross-species diversity. Each sequence is sampled in forward and reverse-complement orientations with equal probability; as noted in Section 3.2.1, this data-level augmentation supplies the biological RC symmetry that parameter tying alone does not provide.

Complete training details are provided in Appendix A, with ablation studies in Appendix B.

### 3.6. Downstream Evaluation

We evaluate LDARNet on two benchmark suites: the **Nucleotide Transformer (NT) tasks** (Dalla-Torre et al., 2025)

with 18 datasets spanning histone modifications, regulatory elements, and splice sites, and **Genomic Benchmarks (GB)** (Grešová et al., 2023) with 9 classification tasks focused on regulatory genomics.

**Evaluation setup.** We follow the protocol of Generator (Wu et al., 2025): for each task, we perform an exhaustive hyperparameter search over 9 learning rates and 4 batch sizes (36 configurations), select the best configuration on validation data, and report test metrics from 10-fold cross-validation. We report Matthews Correlation Coefficient (MCC) for NT tasks and top-1 accuracy for GB tasks, as mean $\pm$ standard deviation across folds. All baseline numbers in Tables 1 and 2 are taken from Wu et al. (2025), which applies this protocol to every model; we re-ran only the LDARNet fine-tuning under the same protocol. Details are in Appendix A.4.

**Models compared.** We benchmark LDARNet against state-of-the-art genomic foundation models, grouping them by scale. **Compact models** (<300M parameters): Enformer (252M) (Avsec et al., 2021), DNABERT-2 (117M) (Zhou et al., 2023), HyenaDNA (55M) (Nguyen et al., 2023), Caduceus-Ph and Caduceus-PS (8M each) (Schiff et al., 2024), GROVER (87M) (Sanabria et al., 2024), and LDARNet (120M). **Large-scale models** ($\geq$300M): NT-multi (2.5B) and NT-v2 (500M) (Dalla-Torre et al., 2025), and Generator and Generator-All (1.2B each) (Wu et al., 2025). Model details are in Appendix A.6.

## 4. Downstream Results

**Nucleotide Transformer Tasks.** On the 18-task NT benchmark (Table 1), **LDARNet achieves 11 wins among compact models (<300M parameters), versus 2 for the next-best alternatives**. Notably, LDARNet secures 8 of 10 histone modification tasks and establishes the best overall result on 5 (H3K4me1, H3K4me2, H3K4me3, H3K79me3, H4ac), including against models up to 2.5B parameters. This is consistent with the long-range nature of histone modification signals; we examine the mechanism in Section 5. On regulatory element and splice-site prediction, LDARNet attains the best compact-model performance on Enhancer classification (57.7 MCC) while remaining competitive on promoter identification.

**Genomic Benchmarks.** On the 9-task GB suite (Table 2), LDARNet ties with DNABERT-2 for best compact-model performance (3 wins each). GB tasks are characterized by high baseline accuracies (>90-95%), creating a ceiling effect that limits performance differentiation. Nevertheless, **on Human non-TATA Promoters LDARNet achieves 96.3% accuracy – the highest across all evaluated models**, exceeding Generator-1.2B (95.8%) and NT-v2-500M

*Table 1.* **Nucleotide Transformer tasks comparison.** Models are grouped by size: <300M parameters (left) and ≥300M parameters (right). **Bold** indicates the best result overall, underlined indicates the best result among models <300M. Best performing model <300M: LDARNet (11/18 wins). Values shown as mean ± std across folds.

| Task | Enformer 252M | DNABERT-2 117M | HyenaDNA 55M | Caduceus-Ph 8M | Caduceus-PS 8M | GROVER 87M | LDARNet 120M | NT-multi 2.5B | NT-v2 500M | Generator 1.2B | Generator-All 1.2B |
|---|---|---|---|---|---|---|---|---|---|---|---|
| # Wins | 0 | 2 | 2 | 1 | 2 | 1 | **11** | - | - | - | - |
| **H3** | 72.4 ± 1.8 | 78.5 ± 1.2 | 78.1 ± 1.5 | 79.4 ± 1.2 | 77.2 ± 2.2 | 76.8 ± 0.8 | 78.2 ± 1.2 | 79.3 ± 1.3 | 78.8 ± 1.0 | **80.6 ± 0.5** | 80.3 ± 0.7 |
| **H3K14ac** | 28.4 ± 2.4 | 51.5 ± 0.9 | **60.8 ± 2.0** | 56.4 ± 3.3 | 59.6 ± 3.8 | 54.8 ± 2.0 | 58.9 ± 3.6 | 53.8 ± 0.9 | 53.8 ± 1.5 | 60.5 ± 0.8 | 58.0 ± 3.8 |
| **H3K36me3** | 34.5 ± 1.9 | 59.1 ± 0.5 | 61.4 ± 1.4 | 59.0 ± 1.8 | 61.1 ± 4.8 | 56.3 ± 1.7 | 62.4 ± 0.7 | 61.8 ± 1.1 | 61.8 ± 1.5 | **65.7 ± 0.7** | 63.1 ± 1.3 |
| **H3K4me1** | 29.1 ± 1.6 | 51.2 ± 0.8 | 51.2 ± 0.8 | 46.8 ± 1.5 | 48.7 ± 2.9 | 46.1 ± 1.8 | **58.3 ± 1.2** | 54.1 ± 0.5 | 54.4 ± 0.9 | 55.3 ± 0.9 | 54.9 ± 1.8 |
| **H3K4me2** | 20.7 ± 2.1 | 33.3 ± 1.3 | 45.5 ± 2.8 | 33.2 ± 3.4 | 43.1 ± 1.6 | 40.3 ± 4.2 | **49.6 ± 1.4** | 32.4 ± 1.4 | 30.2 ± 2.0 | 42.4 ± 1.3 | 40.0 ± 1.5 |
| **H3K4me3** | 15.6 ± 2.2 | 35.3 ± 2.1 | 55.0 ± 1.5 | 49.0 ± 4.2 | 52.8 ± 3.3 | 45.8 ± 2.2 | **57.6 ± 4.3** | 40.8 ± 1.1 | 43.7 ± 2.8 | 51.2 ± 0.9 | 47.3 ± 4.7 |
| **H3K79me3** | 49.8 ± 1.3 | 61.5 ± 1.0 | 66.9 ± 1.4 | 64.1 ± 2.8 | 68.2 ± 1.8 | 62.6 ± 2.6 | **68.7 ± 2.5** | 62.3 ± 1.0 | 62.1 ± 1.2 | 67.0 ± 1.1 | 63.1 ± 2.1 |
| **H3K9ac** | 41.5 ± 2.0 | 54.5 ± 0.9 | 58.6 ± 2.1 | 57.5 ± 2.4 | 56.4 ± 1.8 | 58.1 ± 1.5 | 60.3 ± 2.1 | 54.7 ± 1.1 | 56.7 ± 2.0 | **61.2 ± 0.6** | 60.3 ± 1.9 |
| **H4** | 73.5 ± 2.3 | 79.7 ± 0.8 | 76.3 ± 1.2 | 78.8 ± 1.0 | 79.9 ± 1.0 | 76.9 ± 1.7 | 81.3 ± 1.1 | 80.8 ± 0.7 | 79.5 ± 0.8 | **81.5 ± 0.8** | 80.8 ± 1.0 |
| **H4ac** | 27.5 ± 2.2 | 46.5 ± 1.3 | 56.4 ± 1.1 | 54.8 ± 2.7 | 58.5 ± 1.8 | 53.0 ± 1.7 | **62.3 ± 1.4** | 49.2 ± 1.4 | 50.2 ± 2.5 | 59.2 ± 1.5 | 56.5 ± 3.5 |
| **Enhancer** | 45.4 ± 2.9 | 52.5 ± 2.6 | 52.0 ± 3.1 | 52.2 ± 2.4 | 51.1 ± 2.6 | 51.6 ± 1.8 | 57.7 ± 1.4 | 54.5 ± 2.8 | 56.1 ± 2.9 | **58.0 ± 1.5** | 54.0 ± 2.6 |
| **Enhancer type** | 31.2 ± 4.3 | 42.3 ± 1.8 | 40.3 ± 5.6 | 40.3 ± 2.8 | 41.0 ± 2.6 | 43.3 ± 2.9 | 42.0 ± 2.7 | 44.4 ± 2.2 | 44.4 ± 3.6 | **47.7 ± 1.7** | 46.3 ± 2.3 |
| **Promoter all** | 91.0 ± 0.4 | 94.5 ± 0.3 | 91.9 ± 0.3 | 93.7 ± 0.2 | 94.1 ± 0.3 | 92.6 ± 0.4 | 94.6 ± 0.3 | 95.1 ± 0.4 | 95.2 ± 0.2 | **96.2 ± 0.2** | 95.5 ± 0.2 |
| **Promoter non-TATA** | 91.0 ± 0.6 | 94.4 ± 0.3 | 91.9 ± 0.4 | 93.5 ± 0.7 | 94.0 ± 0.2 | 92.5 ± 0.6 | 94.4 ± 0.5 | 95.5 ± 0.3 | 95.2 ± 0.3 | **96.2 ± 0.1** | 95.5 ± 0.2 |
| **Promoter TATA** | 92.0 ± 1.2 | 91.1 ± 1.1 | 88.1 ± 2.0 | 89.5 ± 1.0 | 90.3 ± 1.0 | 89.1 ± 0.9 | 92.3 ± 0.5 | 91.9 ± 0.8 | 93.3 ± 0.9 | **94.8 ± 0.8** | 93.1 ± 0.7 |
| **Splice acceptor** | 77.2 ± 0.7 | 90.9 ± 0.4 | 93.5 ± 0.5 | 91.8 ± 1.7 | 90.7 ± 1.5 | 91.2 ± 1.0 | 92.7 ± 0.9 | 97.3 ± 0.2 | 97.3 ± 0.4 | **98.1 ± 0.2** | 95.7 ± 0.9 |
| **Splice site all** | 83.1 ± 1.2 | 95.0 ± 0.3 | 91.7 ± 0.6 | 93.5 ± 1.1 | 95.3 ± 0.5 | 91.9 ± 0.5 | 94.2 ± 1.6 | 97.4 ± 0.4 | 97.5 ± 0.2 | **97.8 ± 0.1** | 97.3 ± 0.2 |
| **Splice donor** | 81.3 ± 1.5 | 92.7 ± 0.3 | 89.4 ± 1.3 | 91.2 ± 0.9 | 93.0 ± 1.0 | 88.8 ± 1.2 | 92.8 ± 1.9 | 97.4 ± 0.2 | 97.7 ± 0.7 | **97.8 ± 0.2** | 96.7 ± 0.5 |

*Table 2.* **Genomic Benchmarks comparison.** Models are grouped by size: <300M parameters (left) and ≥300M parameters (right). **Bold** indicates the best result overall, underlined indicates the best result among models <300M. LDARNet ties with DNABERT-2 for best compact-model performance (3/9 wins each). Values shown as mean ± std across folds.

| Benchmark | DNABERT-2 117M | HyenaDNA 55M | Caduceus-Ph 8M | Caduceus-PS 8M | GROVER 87M | LDARNet 120M | NT-v2 500M | Generator 1.2B | Generator-All 1.2B |
|---|---|---|---|---|---|---|---|---|---|
| # Wins | **3** | 0 | 2 | 2 | 0 | **3** | - | - | - |
| **Coding vs. Intergenomic** | 95.1 ± 0.2 | 90.2 ± 0.4 | 93.3 ± 0.1 | 94.4 ± 0.2 | 91.9 ± 0.2 | 95.5 ± 0.1 | 95.5 ± 0.1 | **96.3 ± 0.0** | 95.9 ± 0.1 |
| **Drosophila Enhancers Stark** | 77.4 ± 1.1 | 77.0 ± 1.6 | **82.7 ± 1.0** | 81.6 ± 1.5 | 76.1 ± 1.1 | 81.0 ± 0.8 | 79.7 ± 0.9 | 82.1 ± 0.5 | 76.8 ± 1.5 |
| **Human Enhancers Cohn** | 75.8 ± 0.5 | 72.5 ± 0.9 | 74.7 ± 0.3 | 74.9 ± 0.3 | 73.8 ± 0.3 | 75.2 ± 0.3 | 75.6 ± 0.6 | **76.3 ± 0.2** | 75.4 ± 0.6 |
| **Human Enhancers Ensembl** | 91.8 ± 0.3 | 90.1 ± 0.3 | **92.4 ± 0.2** | 92.3 ± 0.2 | 91.1 ± 0.4 | 90.6 ± 0.7 | 92.1 ± 0.4 | 91.7 ± 0.2 | 91.2 ± 0.2 |
| **Human Ensembl Regulatory** | 87.4 ± 0.7 | 93.2 ± 0.1 | 93.8 ± 0.4 | **94.1 ± 0.2** | 89.7 ± 0.1 | **94.1 ± 0.1** | **94.1 ± 0.1** | 92.8 ± 0.1 | 92.6 ± 0.1 |
| **Human non-TATA Promoters** | 95.7 ± 0.8 | 89.4 ± 2.3 | 96.1 ± 0.3 | 96.1 ± 0.2 | 95.0 ± 0.5 | **96.3 ± 0.4** | 93.2 ± 0.6 | 95.8 ± 0.1 | 95.5 ± 0.5 |
| **Human OCR Ensembl** | 80.6 ± 0.3 | 77.4 ± 0.4 | 82.5 ± 0.4 | **82.6 ± 0.3** | 78.9 ± 0.2 | 79.8 ± 0.3 | 81.3 ± 0.1 | 82.3 ± 0.2 | 81.2 ± 0.3 |
| **Human vs. Worm** | 97.7 ± 0.1 | 95.8 ± 0.4 | 97.5 ± 0.1 | 97.6 ± 0.1 | 96.6 ± 0.1 | 97.6 ± 0.0 | 97.6 ± 0.1 | **98.0 ± 0.0** | 97.8 ± 0.1 |
| **Mouse Enhancers Ensembl** | 86.5 ± 1.4 | 75.6 ± 3.0 | 78.8 ± 2.8 | 82.6 ± 2.1 | 74.2 ± 2.5 | 78.2 ± 2.6 | 85.5 ± 1.8 | **87.1 ± 1.5** | 78.4 ± 2.7 |

(93.2%). Caduceus models (8M parameters) exhibit competitive GB performance attributable to human-genome-specific training, but this specialization constrains cross-species transfer: only 1–2 NT wins compared to LDARNet's 11, illustrating a trade-off between domain specialization and architectural generality.

**Parameter Efficiency.** With 120M parameters, LDARNet achieves the best overall result on 5 NT tasks despite 4–20× parameter disadvantages relative to 500M–2.5B baselines. The combination of learned hierarchical compression and the BiMamba-2 backbone achieves parity with much larger baselines at substantially reduced compute. The next section identifies which of these choices is responsible.

# 5. Why Learned Boundaries Matter

The previous section establishes that LDARNet matches or exceeds much larger baselines, particularly on histone modification tasks. A natural concern is attribution: is this driven by the hierarchical architecture, by the BiMamba-2 backbone, by the multi-species corpus, or specifically by *learned* adaptive boundaries? We answer this with a FLOPs-matched controlled experiment, and corroborate the answer with a biological interpretability analysis of the learned boundary positions.

## 5.1. FLOPs-Matched Controlled Comparison

To isolate the contribution of the routing module under matched conditions, we use a scaled-down proxy of LDARNet (2.5M parameters, 20,000 steps, batch 128, learning rate $1 \times 10^{-3}$, training corpus of 2.56B tokens) and train two matched baselines: `nochunk`, with no compression – every nucleotide is processed by the backbone, requiring ~4× more backbone FLOPs and serving as a compute upper bound; and `fixbound`, with identical architecture and identical compression ratio $N = 4$, but with boundaries frozen every 4 nucleotides – same parameter count, same training data, *same FLOP budget* as the proxy. The only variable between LDARNet and `fixbound` is learned vs. fixed routing.

Models are evaluated by fine-tuning on a representative 7-task NT subset (4 histone modifications, promoter, enhancer, splice) with 5-fold cross-validation; results in Table 3.

*Table 3.* **Learned vs. fixed boundaries at matched FLOPs (2.5M params,** $N = 4$**).** LDARNet uses learned routing; `fixbound` freezes boundaries every 4 nucleotides at identical parameters and FLOPs; `nochunk` is the no-compression upper bound at $\sim4\times$ FLOPs. Bold marks the best result among FLOPs-matched variants (LDARNet vs. `fixbound`).

| Task | LDARNet ($N = 4$) | `fixbound` ($N = 4$) | `nochunk` ($4\times$ FLOPs) |
|---|---|---|---|
| Promoter | $0.916 \pm 0.003$ | **$0.917 \pm 0.006$** | $0.926 \pm 0.003$ |
| Enhancers | **$0.487 \pm 0.040$** | $0.470 \pm 0.052$ | $0.471 \pm 0.037$ |
| Splice | $0.901 \pm 0.046$ | **$0.961 \pm 0.007$** | $0.898 \pm 0.007$ |
| H3 | **$0.771 \pm 0.014$** | $0.723 \pm 0.017$ | $0.792 \pm 0.013$ |
| H3K4me1 | **$0.500 \pm 0.007$** | $0.357 \pm 0.029$ | $0.529 \pm 0.012$ |
| H3K4me3 | **$0.391 \pm 0.013$** | $0.327 \pm 0.287$ | $0.471 \pm 0.057$ |
| H3K36me3 | **$0.560 \pm 0.005$** | $0.493 \pm 0.063$ | $0.597 \pm 0.006$ |

**Histones: learned routing drives the gain.** At identical compute, LDARNet beats `fixbound` by **+14.3pp on H3K4me1, +6.7pp on H3K36me3, and +4.8pp on H3**. These gains cannot be attributed to capacity, scale, training data, or hierarchical architecture – all four are matched. The only variable that differs is the learned routing module, and the only thing it controls is *where* boundaries are placed. Learning where to compress, rather than compressing on a fixed grid, drives most of the histone gain at matched compute. Beyond the mean differences, `fixbound` also exhibits substantially higher variance on the more difficult histone tasks (e.g., std 0.287 on H3K4me3 vs. 0.013 for LDARNet), indicating that fixed boundaries produce unstable representations on these targets.

**Recovering uncompressed performance.** The `nochunk` baseline only modestly exceeds LDAR-Net (e.g., +0.029 on H3K4me1, +0.037 on H3K36me3): learned compression at $N = 4$ recovers nearly the full uncompressed performance at one quarter of the backbone FLOPs, indicating that LDARNet's learned boundaries preserve the information downstream histone-prediction heads need.

**Splice sites: the trade-off reverses.** On splice-site classification the pattern reverses: `fixbound` (0.961) outperforms LDARNet (0.901) at $N = 4$. Splice donors and acceptors are local 2-nt motifs (GT/AG): fixed boundaries preserve uniform positional information across the sequence, while learned routing is not specifically rewarded for landing on every junction. We characterize this trade-off in Appendix B.2 and discuss it in Limitations.

### 5.2. Boundaries Land on Functional Elements

Having established that learned boundaries drive histone performance, we ask whether those boundaries align with biologically recognizable elements. We extracted boundary probabilities produced by the routing module on held-out genomic sequences containing canonical promoter motifs (TATA-box, $n = 76$; CCAAT-box, $n = 73$; GC-box, $n = 163$) and splice sites (GT donors and AG acceptors,

$n = 80$ each), aligned each sequence to the corresponding motif start or splice junction, and averaged boundary probabilities across the alignment. For splice sites we additionally constructed length- and GC-content-matched non-splice controls.

Figure 2 shows localized increases in boundary probability around canonical promoter motifs. The CCAAT-box produces the strongest signal (peak $> 0.6$), while TATA-box and GC-box show weaker but consistent motif-localized peaks. These enrichments indicate that the router tends to place boundaries near transcription-factor binding motifs, treating such elements as segmentation landmarks rather than splitting them arbitrarily across compressed tokens.

At splice junctions (Figure 3), true donor and acceptor sites exhibit elevated boundary probabilities relative to length- and GC-matched non-splice controls. This indicates that the router treats exon–intron transitions as natural segmentation points, even though it was never trained on splice annotations.

These patterns emerge purely from self-supervised pretraining without motif supervision. Combined with the FLOPs-matched controlled comparison above, these results support a mechanistic interpretation of the learned tokenization: **LDARNet places boundaries near biologically meaningful sequence landmarks rather than at arbitrary positions, consistent with the gains of learned routing on histone tasks**, where signal is concentrated in motif-dense regulatory regions. The hierarchical architecture supplies the scaffolding; learning where to place boundaries on that scaffolding is what makes the compression biologically informed.

## 6. Conclusion

We present **LDARNet**, a 120M-parameter hierarchical genomic foundation model that adapts H-Net-style dynamic chunking from autoregressive generation to masked language modeling through bidirectional routing and a bidirectional EMA dechunker designed to reduce sensitivity to masked positions. Across 27 downstream tasks, LDAR-Net achieves 11/18 wins on NT among compact models ($<300M$ parameters) and the best overall performance on 5 histone modification tasks, surpassing models up to $20\times$ larger. A FLOPs-matched controlled experiment isolates the contribution of learned routing (+14.3pp on H3K4me1 over fixed boundaries at identical compute), and nucleotide-resolution analysis shows that learned boundaries are enriched near canonical promoter motifs and splice junctions without supervision.

These results yield two findings of broader interest. First, the benefit of learned tokenization is *task-dependent rather than uniform*: learned routing dominates on long-range

**Figure R2: Learned boundaries around promoter motifs**



*Figure 2.* **Boundary probabilities around canonical promoter motifs.** Mean boundary probability with 95% confidence intervals, aligned to motif start positions. Boundary probability is enriched around motif positions, with the strongest effect observed for the CCAAT-box (peak > 0.6).

**Figure R1: Learned boundary profiles at splice sites**

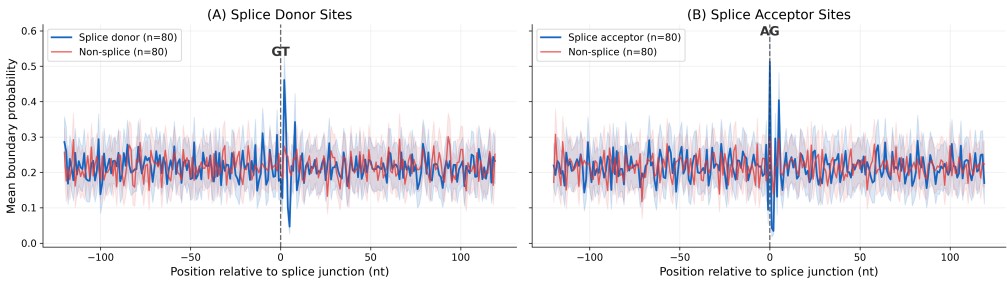

*Figure 3.* **Boundary probabilities at splice junctions.** Mean boundary probability at true donor (GT) and acceptor (AG) sites compared against length- and GC-matched non-splice controls. True junctions show elevated boundary probability relative to controls, indicating that the router identifies exon–intron transitions as segmentation landmarks.

epigenetic tasks, while fixed-grid boundaries can match or exceed it on locally encoded motifs such as splice junctions – a trade-off tunable through the compression ratio. Second, adaptive compression supports strong performance across both cross-species (NT) and human-centric (GB) benchmarks, suggesting that learnable segmentation is a useful inductive bias beyond a single dataset family. Together, these findings indicate that progress in genomic foundation models can come not only from parameter scaling, but from adaptive representations whose structure is itself biologically interpretable.

Natural extensions include multi-stage hierarchical compression for ultra-long contexts (>100kb), zero-shot evaluation of learned representations, and multimodal integration with orthogonal genomic measurements such as RNA-seq, ATAC-seq, and Hi-C.

# 7. Limitations

While LDARNet achieves strong performance among compact genomic foundation models, several limitations warrant discussion.

**Splice-site performance.** Large-scale models (Generator-1.2B, NT-v2-500M) retain advantages on splice-site prediction, where the relevant signal is highly localized (2-nt GT/AG motifs) and the learned router is not specifically rewarded for placing a boundary on every junction. The compression-ratio ablation in Appendix B.2 shows that this gap is partially recoverable at higher $N$ (e.g. splice MCC rises from 0.901 at $N = 4$ to 0.946 at $N = 8$), but not eliminated and at the cost of histone performance.

**Single-stage compression.** Our implementation employs single-stage $4\times$ compression. Multi-stage hierarchies would enable ultra-long context (>100kb) relevant for chromatin interactions and structural variants, but require careful design of gradient flow across compression stages.

**Evaluation scope.** Our assessment focuses on supervised fine-tuning for classification tasks. Zero-shot and few-shot protocols would provide complementary characterization of representation transferability. Pretraining used sequences of up to 4096 bp; systematic evaluation on longer contexts would better characterize the efficiency advantages of hierarchical compression at scale.

**Scope of interpretability analysis.** Our motif-aligned boundary analysis demonstrates that learned segmentation recovers canonical promoter and splice elements, but a systematic genome-scale correspondence with experimentally validated features (e.g., ChIP-seq peaks, full TF binding catalogs) remains future work.

## Acknowledgements

This work was supported by the Ministry of Economic Development of the Russian Federation (agreement No. 139-15-2025-013, dated June 20, 2025, subsidy identifier 000000C313925P4B0002).

## Impact Statement

LDARNet demonstrates that compact, parameter-efficient genomic foundation models can match much larger alternatives on a broad set of regulatory and epigenetic prediction tasks. This has two downstream implications. First, it reduces the compute required for genomic representation learning, making such models accessible to research groups without large-scale infrastructure. Second, the interpretability of the learned tokenization – segmentation boundaries that align with canonical promoter and splice motifs – offers a path toward genomic models whose internal structure can be inspected against known biology, which is relevant for downstream applications in clinical and regulatory settings where black-box predictions are problematic.

All experiments use publicly available benchmarks and do not involve private, identifiable, or clinical human data. As with other advances in genomic modeling, improvements in DNA sequence understanding could in principle accelerate misuse scenarios such as the engineering of pathogenic sequences; however, LDARNet operates on the level of regulatory classification and does not introduce new generative capabilities relevant to such risks. We see no specific harms uniquely enabled by this work.

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

# A. Training Details

## A.1. Architecture Specification

LDARNet is instantiated with approximately 120M parameters following the hierarchical layout `[m3t1, [M10], m4]`. The encoder comprises three BiMamba-2 layers followed by a single local attention layer; the backbone consists of ten BiMamba-2 layers operating in the compressed latent space; and the decoder is composed of four BiMamba-2 layers that restore the sequence to its original resolution. This asymmetric allocation prioritizes efficient sequence processing at full resolution in the outer stages while concentrating representational capacity in the compressed backbone, where each latent token aggregates information from a variable-length span of the original sequence.

Model dimensions follow the H-Net convention of widening compressed stages (Hwang et al., 2025) to compensate for the reduced number of tokens after chunking. The outer stages (encoder and decoder) operate at $d_{\text{model}} = 512$, while the backbone uses $d_{\text{model}} = 768$ with a feed-forward hidden dimension of $d_{\text{ff}} = 2560$. BiMamba-2 state-space modules are configured with chunk size 256, convolutional kernel width 4, state dimension $d_{\text{state}} = 128$, and an expansion factor of 2. The local attention layer in the encoder uses 16 heads with rotary position embeddings of dimension 32 and a local window of size 1023, providing a complementary short-range inductive bias for fine-grained motif recognition before compression.

The input vocabulary consists of seven byte-level tokens $\{A, C, G, T, N, \texttt{[PAD]}, \texttt{[MASK]}\}$, with untied input and output embeddings to allow asymmetric optimization of the reconstruction objective.

## A.2. Training Objective and Optimization

The pretraining objective combines masked language modeling with compression regularization:

$$\mathcal{L} = \mathcal{L}_{\text{MLM}} + \alpha \sum_{s=0}^{S-1} \mathcal{L}_{\text{ratio}}^s, \tag{18}$$

where $\mathcal{L}_{\text{MLM}}$ is the standard reconstruction loss over masked positions and $\mathcal{L}_{\text{ratio}}^s$ is the ratio-based regularizer at stage $s$ (Section 3.4), which encourages the router to select approximately a fraction $1/N$ of positions as boundaries. For the single-stage LDARNet ($S = 1$), this reduces to a single ratio term. We set $\alpha = 0.03$ to balance reconstruction quality with adherence to the target compression ratio, and apply token masking with probability 15% during training.

Models are optimized with AdamW (Loshchilov & Hutter, 2017) using $\beta_1 = 0.9$, $\beta_2 = 0.95$ (following Dao & Gu (2024)), $\epsilon = 10^{-8}$, and weight decay 0.01. Gradients are clipped to a maximum norm of 1.0. The learning rate follows a warmup–stable–decay (WSD) schedule with base rate $5 \times 10^{-4}$: linear warmup over the first 10% of training, a constant plateau for 70%, and inverse-square-root decay over the final 20% ending at 5% of the peak value.

Following prior work on hierarchical models (Hwang et al., 2025), we apply stage-wise learning rate scaling to account for gradient-magnitude differences arising from the compressed backbone. Outer-stage parameters (encoder and decoder) receive

$$\eta_{\text{outer}} = \eta_{\text{base}} \cdot \sqrt{N} \cdot \frac{d_{\text{back}}}{d_{\text{outer}}}, \tag{19}$$

which compensates for the gradient attenuation introduced by the chunking and dechunking operations between resolutions. For our configuration ($N = 4$, $d_{\text{outer}} = 512$, $d_{\text{back}} = 768$), this yields a $3\times$ multiplier for outer layers relative to the backbone.

Training uses gradient accumulation over 16 steps with micro-batch size 32, giving an effective batch size of 512 sequences per iteration. All input sequences are of length 4096 nucleotides.

## A.3. Training Infrastructure

Training is performed with PyTorch DistributedDataParallel (DDP) over the NCCL backend across multiple GPUs. Mixed-precision training uses `bfloat16` throughout, and we enable TF32 matrix operations on Ampere-generation GPUs for additional throughput.

The pretraining corpus combines the human reference genome (GRCh38/hg38) with the multispecies collection from Nucleotide Transformer (Dalla-Torre et al., 2025), ensuring both in-species fidelity and cross-species diversity. Genomic intervals are sampled uniformly across the corpus and sharded across workers to ensure even coverage during training. To

encourage reverse-complement invariance at the data level, each sequence is sampled in forward and reverse-complement orientations with equal probability.

## A.4. Downstream Evaluation Setup

We follow the evaluation protocol of Generator (Wu et al., 2025), fine-tuning LDARNet on each benchmark task with 10-fold cross-validation. For every (model, dataset) pair, we perform an exhaustive hyperparameter search over learning rates in $\{1e-5, 2e-5, 5e-5, 1e-4, 2e-4, 5e-4, 1e-3, 2e-3, 5e-3\}$ and batch sizes in $\{64, 128, 256, 512\}$, yielding 36 configurations per task. Early stopping is applied based on validation loss with patience of 5 epochs. The baseline numbers reported in Tables 1 and 2 were obtained under the same protocol in Wu et al. (2025), ensuring a directly comparable evaluation across all models.

For each task, we select the hyperparameter configuration that achieves the best validation performance, then report test metrics averaged over the cross-validation folds together with the standard deviation across folds. This protocol mitigates confounds from suboptimal hyperparameter selection and provides a stable estimate of test-time performance.

## A.5. Optimal Hyperparameters per Task

Tables 4 and 5 report the learning rate and effective batch size selected by the hyperparameter search described in Section A.4. These configurations were then used for the 10-fold cross-validation results reported in Tables 1 and 2.

*Table 4.* **Optimal hyperparameters for Nucleotide Transformer tasks.** Learning rate (LR) and effective batch size (BS) selected by validation performance from 36 configurations per task.

| Task | Learning Rate | Batch Size |
|------|---------------|------------|
| *Histone Modifications* | | |
| H3 | $1 \times 10^{-3}$ | 64 |
| H3K4me1 | $5 \times 10^{-3}$ | 128 |
| H3K4me2 | $5 \times 10^{-3}$ | 512 |
| H3K4me3 | $1 \times 10^{-3}$ | 64 |
| H3K9ac | $1 \times 10^{-3}$ | 128 |
| H3K14ac | $1 \times 10^{-3}$ | 128 |
| H3K36me3 | $5 \times 10^{-4}$ | 64 |
| H3K79me3 | $2 \times 10^{-3}$ | 128 |
| H4 | $2 \times 10^{-3}$ | 64 |
| H4ac | $1 \times 10^{-3}$ | 64 |
| *Regulatory Elements* | | |
| Enhancer | $2 \times 10^{-4}$ | 128 |
| Enhancer type | $2 \times 10^{-3}$ | 64 |
| Promoter all | $1 \times 10^{-3}$ | 128 |
| Promoter non-TATA | $5 \times 10^{-3}$ | 256 |
| Promoter TATA | $1 \times 10^{-3}$ | 128 |
| *Splice Sites* | | |
| Splice acceptor | $1 \times 10^{-3}$ | 128 |
| Splice donor | $2 \times 10^{-3}$ | 64 |
| Splice site all | $5 \times 10^{-3}$ | 128 |

*Table 5.* **Optimal hyperparameters for Genomic Benchmarks tasks.** Learning rate (LR) and effective batch size (BS) selected by validation performance from 36 configurations per task.

| Task | Learning Rate | Batch Size |
|------|:---:|:---:|
| Coding vs. Intergenomic | $1 \times 10^{-3}$ | 64 |
| Drosophila Enhancers Stark | $2 \times 10^{-3}$ | 128 |
| Human Enhancers Cohn | $2 \times 10^{-3}$ | 64 |
| Human Enhancers Ensembl | $1 \times 10^{-3}$ | 128 |
| Human Ensembl Regulatory | $2 \times 10^{-3}$ | 64 |
| Human non-TATA Promoters | $1 \times 10^{-3}$ | 128 |
| Human OCR Ensembl | $2 \times 10^{-3}$ | 64 |
| Human vs. Worm | $1 \times 10^{-3}$ | 64 |
| Mouse Enhancers Ensembl | $1 \times 10^{-3}$ | 128 |

### A.6. Baseline Model Architectures

This section summarizes the key architectural and training characteristics of all baseline models evaluated in our downstream benchmarks.

**Compact Models ($<$ 300M parameters).** **Enformer (252M)** (Avsec et al., 2021) is a Transformer-based model trained in a supervised multi-task fashion to predict thousands of chromatin and gene expression tracks (from ENCODE and Roadmap Epigenomics) directly from DNA sequence. While the other compact baselines use self-supervised pretraining objectives (MLM or next-token prediction), Enformer is used here as a frozen-torso pretrained encoder following the convention of Dalla-Torre et al. (2025). The model processes sequences up to $\sim$196 kb via a convolutional stem feeding Transformer blocks with custom relative positional basis functions designed for genomic context.

**DNABERT-2 (117M)** (Zhou et al., 2023) employs Byte-Pair Encoding (BPE) tokenization with a learned vocabulary, combined with Attention with Linear Biases (ALiBi) positional encoding for extrapolation to longer sequences. The model was pretrained on a multi-species genome corpus using masked language modeling. BPE produces variable-length subword units at the cost of single-nucleotide resolution.

**HyenaDNA (55M)** (Nguyen et al., 2023) replaces self-attention with implicit long convolutions inspired by state-space models, enabling subquadratic processing of very long sequences (up to 1 Mb during pretraining). It uses single-nucleotide tokenization and is pretrained on the human reference genome.

**Caduceus-Ph and Caduceus-PS (8M each)** (Schiff et al., 2024) are bidirectional Mamba-based DNA language models – the smallest baselines in our comparison by a substantial margin. Both variants use single-nucleotide tokenization and are pretrained exclusively on the human reference genome (GRCh38) with the masked language modeling objective. **Caduceus-PS** (parameter sharing) enforces reverse-complement (RC) equivariance *architecturally* via the MambaDNA block, in which parameters are shared between forward and RC channel splits; this removes the need for RC data augmentation during pretraining. **Caduceus-Ph** (post hoc) is a stack of BiMamba blocks without architectural RC equivariance: it is instead pretrained with RC data augmentation, and downstream predictions are obtained by averaging the model outputs on the original sequence and its reverse complement (post-hoc conjoining).

**GROVER (87M)** (Sanabria et al., 2024) is a BERT-style Transformer pretrained with masked language modeling on the human reference genome. Its Byte-Pair Encoding (BPE) vocabulary is constructed via 600 BPE merge cycles on the human genome, with the number of merges selected by a downstream next-$k$-mer prediction evaluation.

**Large-Scale Models ($\geq$300M parameters).** **NT-multi (2.5B) and NT-v2 (500M)** (Dalla-Torre et al., 2025) are encoder-only Transformer models pretrained with masked language modeling on a multi-species genome corpus spanning hundreds of species. Both use 6-mer tokenization with a fallback to single nucleotides for residual positions. NT-multi belongs to the first generation and uses learned positional embeddings, while NT-v2 (the second generation) introduces rotary position embeddings (RoPE) and Gated Linear Units, achieving performance comparable to NT-multi at $5\times$ fewer parameters.

**GENERator and GENERator-All (1.2B each)** (Wu et al., 2025) are autoregressive Transformer decoders pretrained with

next-token prediction over 6-mer tokens, supporting context lengths of up to 98 kb. GENERator is pretrained on $\sim$386 B nucleotides of eukaryotic DNA; GENERator-All is pretrained on an expanded corpus described in the original paper. The two variants otherwise share architecture and training procedure.

### A.7. Detailed Results Analysis

**Nucleotide Transformer Tasks: Histone Modifications.**    The NT benchmark includes 10 histone modification prediction tasks, which probe long-range chromatin context. LDARNet achieves strong performance on this category, winning 8 of 10 tasks among compact models and the best overall result on 5: H3K4me1 (58.3), H3K4me2 (49.6), H3K4me3 (57.6), H3K79me3 (68.7), and H4ac (62.3). On these five tasks, LDARNet surpasses all baselines, including models with up to 20× more parameters (NT-multi at 2.5B). The consistent advantage on the H3K4 methylation family (me1, me2, me3) is consistent with the boundary-interpretability analysis in Section 5.2: H3K4 marks are associated with active promoters and enhancers, and the learned router places boundaries with elevated probability around canonical promoter motifs, suggesting that LDARNet represents promoter regions as coherent units that downstream histone-prediction heads can exploit.

On H3K9ac (60.3) and H4 (81.3), LDARNet attains the best compact-model result and remains within 1 MCC of the best large-scale model (Generator: 61.2 and 81.5, respectively). On H3K36me3, LDARNet (62.4) again leads among compact models, though here the gap to Generator (65.7) is more substantial. The only histone task where LDARNet does not lead among compact models is H3K14ac, on which HyenaDNA achieves 60.8 MCC – the best overall result across all models – despite being roughly half the size of LDARNet.

**Nucleotide Transformer Tasks: Regulatory Elements and Splice Sites.**    On regulatory element prediction, LDARNet attains the best compact-model result on Enhancer (57.7 MCC), within 0.3 MCC of the best large-scale model Generator (58.0). On promoter classification, DNABERT-2 leads compact models on Promoter all (94.5) and ties with LDARNet on Promoter non-TATA (94.4), with Generator achieving the overall best on both. The strength of DNABERT-2 on promoter tasks is likely related to its BPE tokenization, which produces variable-length subwords that align with frequent promoter elements.

Splice-site prediction tasks (Splice acceptor, Splice donor, Splice site all) are dominated by large-scale models: Generator achieves above 97.5 MCC on all three. Among compact models, Caduceus-PS leads on Splice site all (95.3) and Splice donor (93.0), and HyenaDNA leads on Splice acceptor (93.5). The pattern aligns with the splice-site observation in Section 5.1: splice sites are characterized by short, highly localized motifs (GT/AG dinucleotides), where models that preserve uniform single-nucleotide positional information have an advantage over learned hierarchical compression.

**Genomic Benchmarks: Saturation Effects and Specialized Models.**    GB tasks present a different challenge than NT tasks due to performance saturation: on 5 of 9 tasks, the best-performing model exceeds 92% accuracy, and on 3 tasks even compact models exceed 95%. The narrow margins limit the resolution at which architectural differences can be measured.

Nevertheless, LDARNet achieves 3 wins among compact models. On Coding vs. Intergenomic (95.5%), LDARNet leads the compact category, matching the much larger NT-v2 (500M) and approaching Generator (96.3%). On Human Ensembl Regulatory (94.1%), LDARNet ties for the overall best result with Caduceus-PS (compact) and NT-v2 (large-scale), demonstrating that a compact model can reach the same ceiling as a 4× larger one in this regime. On Human non-TATA Promoters (96.3%), LDARNet achieves the single best result across all evaluated models, exceeding Generator (95.8%) and NT-v2 (93.2%).

Caduceus models achieve strong GB performance despite being 15× smaller than LDARNet, with 2 underlined wins each and 3 overall best results across the two variants (Drosophila Enhancers Stark: 82.7%, Human Enhancers Ensembl: 92.4%, Human OCR Ensembl: 82.6%). This performance likely reflects their exclusive pretraining on the human reference genome – they are effectively human-specialist models. The same specialization limits cross-species generalization: on NT, which includes histone tasks derived from yeast, Caduceus-Ph and Caduceus-PS achieve only 1 and 2 wins respectively, compared to LDARNet's 11.

DNABERT-2 shows balanced performance across both benchmarks, with 2 NT wins and 3 GB wins. Its GB wins come on Human Enhancers Cohn, Human vs. Worm, and Mouse Enhancers Ensembl – a mix of cross-species and enhancer-like tasks consistent with the generalist effect of multi-species BPE pretraining.

**Performance-Parameter Efficiency Analysis.** LDARNet (120M) matches or exceeds models with 500M–2.5B parameters on multiple task categories. On the 5 NT tasks where it achieves the best overall result, the closest competitors are $4\times$ (NT-v2, 500M) to $20\times$ (NT-multi, 2.5B) larger. This efficiency is consistent with our central design hypothesis – that learnable hierarchical compression substitutes for raw parameter scale on tasks dominated by long-range structure – and is examined causally in Section 5.1.

The contrast between Caduceus and LDARNet is methodologically informative. Caduceus (8M) achieves strong GB performance through specialization on the human reference genome, while LDARNet (120M) achieves strong performance across both human-centric GB and the multi-species NT benchmark through architectural generality. These represent two distinct routes to parameter efficiency in genomic foundation models: domain specialization, and architectural innovation under broader pretraining. The latter route is the focus of this work and is more directly aligned with general-purpose downstream use.

### A.8. Computational Budget

**Pretraining.** LDARNet (120M parameters) was pretrained on $6\times$ NVIDIA A100 80GB GPUs with mixed-precision (`bfloat16`). The pretraining corpus combines the human reference genome (GRCh38) with the multi-species collection from Nucleotide Transformer (Dalla-Torre et al., 2025) ($\sim$300 B base pairs). Training used sequences of length 4096 with effective batch size 512 and required 7 days of wall-clock time, totaling **1,008 GPU-hours (42 GPU-days)**.

**Downstream Evaluation.** For each task, we performed an exhaustive hyperparameter search over 36 configurations (9 learning rates $\times$ 4 batch sizes), followed by 10-fold cross-validation with the optimal configuration. Per-configuration training time ranged from 30 minutes to 6 hours depending on task complexity and dataset size, with a mean of roughly 2 hours. This yields approximately $27 \times 36 \times 2 = 1,944$ GPU-hours (81 GPU-days) for the hyperparameter search and $27 \times 10 \times 2 = 540$ GPU-hours (22.5 GPU-days) for cross-validation, summing to **2,484 GPU-hours (103.5 GPU-days)** of downstream compute.

**Total Cost.** The complete experimental pipeline used **3,492 GPU-hours ($\approx$145.5 GPU-days)** on A100 80GB hardware. For context, pretraining a $\sim$2.5B-parameter model at the scale of NT-multi is estimated to require on the order of $10^3$ GPU-days; LDARNet's full pipeline (pretraining + all downstream experiments) thus uses roughly an order of magnitude less compute than a single training run at that scale. Assuming 400 W per A100 and a datacenter PUE of 1.2, total energy consumption is approximately 1,680 kWh, corresponding to an estimated $\sim$670 kg $CO_2$e at a typical grid carbon intensity of 0.4 kg $CO_2$/kWh, following the reporting framework of Patterson et al. (2021).

## B. Ablation Studies

To isolate the contribution of individual design choices, we run a controlled ablation suite using 2.5M-parameter proxy models trained under identical conditions. All variants are trained for 20,000 steps on the same corpus (2.56B tokens) with batch size 128 and learning rate $1 \times 10^{-3}$. Downstream evaluation uses 7 representative NT tasks spanning histone modifications (H3, H3K4me1, H3K4me3, H3K36me3), regulatory elements (Promoter, Enhancers), and Splice sites. For each task we perform 5-fold cross-validation under a fixed-hyperparameter fine-tuning protocol (batch size 128, learning rate $1 \times 10^{-3}$, early stopping with patience 5 epochs), and report MCC averaged over folds with standard deviation. This protocol enables direct, FLOPs-matched comparison of design choices at controlled compute.

### B.1. Architecture Layout

We vary the placement of Mamba-2 and Transformer blocks across the encoder, backbone, and decoder, holding compression ratio ($N = 4$), fusion (mean), and ratio-loss weight ($\alpha = 0.03$) fixed (Table 6). The notation `X-Y-Z` denotes block types in (encoder / backbone / decoder), where `m` = BiMamba-2 outer block, `M` = BiMamba-2 backbone, `t`/`T` = Transformer (outer/backbone), and `mt` = encoder combining BiMamba-2 with a single local attention layer.

Two trends emerge. First, the BiMamba-2 backbone (`*-M-*` configurations) is preferred on three of the four histone tasks (H3, H3K4me1, H3K36me3), while a Transformer backbone (`*-T-*`) wins on splice-site classification by a substantial margin (best 0.964 vs. 0.906) but underperforms on histones – a backbone-driven trade-off we resolve in favor of long-range epigenetic modeling, where LDARNet's improvements over larger baselines are most pronounced. Second, comparing `mt-M-m` against `m-M-m` isolates the role of the single local attention layer in the encoder: it improves H3K4me1 (0.500

*Table 6.* **Architecture layout ablation** (2.5M params, $N = 4$, mean fusion, $\alpha = 0.03$). Notation `X-Y-Z`: encoder / backbone / decoder, where `m` = BiMamba-2 outer block, `M` = BiMamba-2 backbone, `t`/`T` = Transformer (outer/backbone), `mt` = BiMamba-2 plus one local attention layer. Bold marks the best result per column.

| Config | Promoter | Enhancers | Splice | H3 | H3K4me1 | H3K4me3 | H3K36me3 |
|---|---|---|---|---|---|---|---|
| `mt-M-m` | $0.916 \pm 0.003$ | $0.487 \pm 0.040$ | $0.901 \pm 0.046$ | $0.771 \pm 0.014$ | $\mathbf{0.500 \pm 0.007}$ | $0.391 \pm 0.013$ | $\mathbf{0.560 \pm 0.005}$ |
| `m-M-m` | $0.919 \pm 0.006$ | $\mathbf{0.495 \pm 0.033}$ | $0.864 \pm 0.013$ | $\mathbf{0.773 \pm 0.009}$ | $0.448 \pm 0.009$ | $0.420 \pm 0.052$ | $0.521 \pm 0.011$ |
| `m-T-m` | $\mathbf{0.922 \pm 0.003}$ | $0.492 \pm 0.023$ | $0.881 \pm 0.011$ | $0.750 \pm 0.015$ | $0.452 \pm 0.013$ | $\mathbf{0.428 \pm 0.015}$ | $0.519 \pm 0.024$ |
| `mt-T-m` | $0.914 \pm 0.007$ | $0.487 \pm 0.034$ | $\mathbf{0.964 \pm 0.009}$ | $0.750 \pm 0.008$ | $0.495 \pm 0.008$ | $0.398 \pm 0.039$ | $0.555 \pm 0.009$ |
| `t-M-t` | $0.900 \pm 0.005$ | $0.480 \pm 0.013$ | $0.906 \pm 0.127$ | $0.733 \pm 0.007$ | $0.439 \pm 0.018$ | $0.374 \pm 0.043$ | $0.512 \pm 0.004$ |
| `t-T-t` | $0.904 \pm 0.005$ | $0.458 \pm 0.043$ | $0.931 \pm 0.031$ | $0.752 \pm 0.009$ | $0.434 \pm 0.014$ | $0.306 \pm 0.054$ | $0.506 \pm 0.011$ |

*Table 7.* **Compression ratio ablation:** $N$ **at fixed architecture** `mt-M-m`**, mean fusion.** 2.5M-parameter models, fine-tuned with 5-fold cross-validation. Higher $N$ favors splice sites; lower $N$ favors histones. Bold marks the best result per column.

| $N$ | Promoter | Enhancers | Splice | H3 | H3K4me1 | H3K4me3 | H3K36me3 |
|---|---|---|---|---|---|---|---|
| 2 | $\mathbf{0.928 \pm 0.004}$ | $0.491 \pm 0.031$ | $0.909 \pm 0.030$ | $\mathbf{0.776 \pm 0.012}$ | $\mathbf{0.517 \pm 0.008}$ | $\mathbf{0.415 \pm 0.044}$ | $\mathbf{0.598 \pm 0.005}$ |
| 4 | $0.916 \pm 0.003$ | $0.487 \pm 0.040$ | $0.901 \pm 0.046$ | $0.771 \pm 0.014$ | $0.500 \pm 0.007$ | $0.391 \pm 0.013$ | $0.560 \pm 0.005$ |
| 6 | $0.916 \pm 0.005$ | $0.483 \pm 0.026$ | $0.940 \pm 0.009$ | $0.767 \pm 0.009$ | $0.502 \pm 0.018$ | $0.380 \pm 0.028$ | $0.564 \pm 0.009$ |
| 8 | $0.910 \pm 0.005$ | $0.472 \pm 0.033$ | $\mathbf{0.946 \pm 0.007}$ | $0.765 \pm 0.014$ | $0.472 \pm 0.006$ | $0.355 \pm 0.009$ | $0.555 \pm 0.006$ |
| 12 | $0.908 \pm 0.003$ | $\mathbf{0.493 \pm 0.024}$ | $0.915 \pm 0.056$ | $0.759 \pm 0.010$ | $0.477 \pm 0.012$ | $0.375 \pm 0.033$ | $0.541 \pm 0.008$ |

vs. 0.448) and H3K36me3 (0.560 vs. 0.521) – the marks where LDARNet's main-text gains are largest – while slightly reducing performance on H3K4me3 and on regulatory tasks. We adopt the attention layer because the gains align with the targets that drive LDARNet's main-text result. Configurations with attention-only outer modules (`t-T-t` and `t-M-t`) trend weakest on the histone tasks, supporting the choice of BiMamba-2 for the outer stages.

We adopt `mt-M-m` for the main model: the BiMamba-2 backbone (`M`) dominates the histone tasks central to LDARNet's competitive position; the local attention layer in the encoder (`mt`) improves H3K4me1 and H3K36me3, the marks where LDARNet's main-text gains are largest; and BiMamba-2 outer modules outperform attention-only outer modules, justifying `m` for the decoder.

## B.2. Compression Ratio

We sweep the compression ratio $N \in \{2, 4, 6, 8, 12\}$ at fixed architecture (`mt-M-m`, mean fusion, $\alpha = 0.03$). Table 7 reveals a task-dependent trade-off: $N = 2$ achieves the strongest histone performance (best on all four histone tasks), while $N = 8$ produces the largest gains on splice-site classification (Splice MCC 0.946 at $N = 8$ vs. 0.901 at $N = 4$). Promoter accuracy decreases with $N$, and enhancer classification is largely insensitive (variation within one standard deviation across all $N$).

We adopt $N = 4$ for the main model as a compromise between histone and splice-site performance under a reasonable compression budget: it retains most of the histone performance attainable at $N = 2$ while doubling the sequence-length reduction and remaining within 0.05 MCC of the splice-site result at $N = 8$. The $N = 8$ configuration represents an alternative for applications prioritizing splice-site performance, with a corresponding decrease in histone metrics (Section 7).

## B.3. Bidirectional Fusion Strategy

BiMamba-2 combines forward and reverse Mamba-2 passes; we evaluate two parameter-free fusion strategies for combining their outputs: mean fusion (Eq. 6) and sum fusion. The choice produces a task-dependent effect at fixed architecture and compression (Table 8). Mean fusion is stronger on the H3K4 methylation family (H3K4me1: 0.500 vs. 0.477; H3K4me3: 0.391 vs. 0.338) and on H3K36me3 (0.560 vs. 0.547), while sum fusion is stronger on H3 (0.778 vs. 0.771), Enhancers, Promoter, and most notably on splice-site classification (0.941 vs. 0.901).

We adopt mean fusion in the main model because the histone gains (and in particular the H3K4 methylation marks) align with the tasks where LDARNet achieves its largest absolute improvements over much larger baselines in Table 1. Sum fusion could be preferable for splice-focused applications.

*Table 8.* **Bidirectional fusion strategy** (2.5M params, mt-M-m, $N = 4$). Mean fusion favors H3K4 methylation marks; sum fusion favors splice sites and H3. Bold marks the best result per column.

| Fusion | Promoter | Enhancers | Splice | H3 | H3K4me1 | H3K4me3 | H3K36me3 |
|--------|----------|-----------|--------|-----|---------|---------|----------|
| mean | $0.916 \pm 0.003$ | $0.487 \pm 0.040$ | $0.901 \pm 0.046$ | $0.771 \pm 0.014$ | $\mathbf{0.500 \pm 0.007}$ | $\mathbf{0.391 \pm 0.013}$ | $\mathbf{0.560 \pm 0.005}$ |
| sum | $\mathbf{0.919 \pm 0.006}$ | $\mathbf{0.497 \pm 0.058}$ | $\mathbf{0.941 \pm 0.006}$ | $\mathbf{0.778 \pm 0.010}$ | $0.477 \pm 0.009$ | $0.338 \pm 0.013$ | $0.547 \pm 0.008$ |

### B.4. Ratio Loss Weight

The ratio loss, weighted by $\alpha$, regularizes the learned compression toward the target ratio $N$. To verify that this term is both necessary and compatible with the main objective, we compare two configurations differing only in $\alpha$: the main setting $\alpha = 0.03$ and a control with $\alpha = 0$ (regularizer disabled). All other settings are held fixed: mt-M-m architecture, $N = 4$, mean fusion, and identical optimizer, learning rate, batch size, and pretraining corpus. Training dynamics over 20,000 steps are shown in Figure 4.

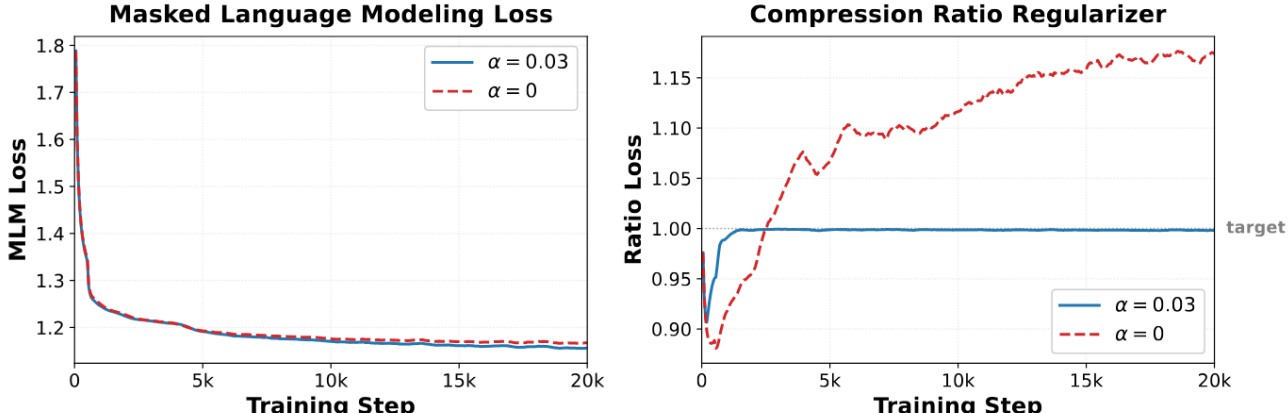

*Figure 4.* **Training dynamics with and without ratio regularization.** Blue: main configuration ($\alpha = 0.03$); red dashed: control ($\alpha = 0$). (a) MLM loss $\mathcal{L}_{\mathrm{MLM}}$: trajectories are nearly indistinguishable, indicating that the regularizer does not interfere with reconstruction. (b) Ratio loss $\mathcal{L}_{\mathrm{ratio}}$: the dashed horizontal line marks the theoretical minimum at $F = G = 1/N$. With $\alpha = 0.03$, $\mathcal{L}_{\mathrm{ratio}}$ converges to its minimum by step $\sim$1500 and remains stable thereafter; without regularization it drifts upward throughout training.

Three observations follow. First, the MLM loss converges to nearly identical values in both configurations – 1.158 at $\alpha = 0.03$ versus 1.169 at $\alpha = 0$ – indicating that ratio regularization imposes no measurable cost on the main reconstruction objective. Second, with $\alpha = 0.03$ the ratio loss approaches its theoretical minimum by step $\sim$1500 and remains stable thereafter (mean $0.998 \pm 0.001$ over the final 1000 steps), showing that the router reliably matches the target compression $N = 4$. Third, without regularization $\mathcal{L}_{\mathrm{ratio}}$ briefly decreases to $\approx 0.87$ before drifting upward to $\approx 1.18$ by the end of training, with no sign of stabilization. The router, optimized only through the MLM gradient, does not naturally maintain the target compression and settles at a different operating point – the ratio $N = 4$ is not favored by the MLM objective alone. The ratio loss is therefore not redundant but a necessary component for controlling the router's compression behavior.

## C. LLM Usage

Large language models (LLMs) were employed to assist with the preparation of this manuscript. Specifically, we used commercial general-purpose LLMs to improve the clarity, coherence, and grammar of the text, and to help rephrase sections for consistency with academic writing standards. All technical content, experimental design, data analysis, and results interpretation were conceived, implemented, and validated by the authors. LLMs were not used to generate novel scientific ideas, design experiments, or analyze results. Final responsibility for the accuracy and integrity of the content rests entirely with the authors.

# D. Reproducibility Statement

All architectural details, training configurations, and evaluation protocols required to reproduce our results are specified in the main text and appendix. Model architecture and hyperparameters are described in Section 3.5, with complete training details in Appendix A. Downstream evaluation protocols, including the hyperparameter search and cross-validation procedure, are detailed in Section 3.6 and Appendix A.4; the per-task optimal configurations are reported in Appendix A.5, and the computational budget is documented in Appendix A.8. All experiments were conducted on publicly available benchmark datasets: the Nucleotide Transformer benchmark (Dalla-Torre et al., 2025) and the Genomic Benchmarks suite (Grešová et al., 2023). Code and pretrained model weights will be made available at `https://github.com/darlednik/ICML-LDARNet`.

