# OpenReview forum: "LDARNet: DNA Adaptive Representation Network with Learnable Tokenization for Genomic Modeling"
_ICML.cc/2026/Conference — ICML 2026 regular_

### Official Review · Reviewer_DXHA · 2026-02-26

**Soundness:** 3
**Presentation:** 2
**Significance:** 3
**Originality:** 3
**Overall Recommendation:** 5
**Confidence:** 3

**Summary:**

LDARNet is a 120M-parameter genomic foundation model that adapts the dynamic chunking mechanism from H-Net to a masked language modeling framework. The model employs a hierarchical architecture, designing BiMamba-2 blocks to learn biologically relevant token boundaries. Evaluation on 27 downstream tasks demonstrates that LDARNet offers high parameter efficiency, frequently outperforming significantly larger models (up to 2.5B parameters) especially on histone modification tasks, while staying competitive on others.

**Compliance With Llm Reviewing Policy:**

Affirmed.

**Final Justification:**

I am increasing my score to a 5. The authors submitted an effective rebuttal that addressed my primary concerns. The new interpretability experiments, which demonstrate that the model's token boundaries naturally align with actual biological structures, increase the paper's significance. Additionally, the expanded ablation studies justify the design choices. I also appreciate their transparent explanation regarding the hardware and compute constraints preventing a direct comparison with Evo 2; their rationale is completely understandable, and the existing benchmarks adequately prove the model's efficiency.

**Key Questions For Authors:**

1. Section 3.2.2 describes the backbone as "Transformer" , but Section 3.5 says "BiMamba-2". Which architecture was actually used for the results?
2. Section 3.2.2 implies Nucleotide Transformer (Lopez et al.) uses BPE, but to the best of our knowledge, this model uses k-mers?
3. Since BiMamba-2 is designed for bi-directionality with shared weights, is the data augmentation (sampling reverse strands) strictly necessary?
4. Did you perform analyses on your design choices for the BiMamba-2 block, such as alternatives to "mean fusion" or disabling parameter tying for the projection layers?
5. The per-task hyper-parameter search seems to be unfair if not done for every other models: did you perform this optimization for all models of Tables 1 and 2?

**Limitations:**

yes

**Strengths And Weaknesses:**

**Strengths:**

1. The authors address a core limitation in genomic modeling (fixed tokenizations) by successfully adapting autoregressive dynamic chunking to a MLM framework. The introduction of a custom BiMamba-2 block with shared weights is a principled design choice that effectively handles the non-causal nature of genomic data.
2. The model demonstrates parameter efficiency, achieving state-of-the-art results on histone modification tasks with only 120M parameters. It notably outperforms baselines with up to 2.5B parameters, with 11 out of 18 wins on the Nucleotide Transformer benchmark suite.
3. The authors provide a comprehensive analysis of compression ratios and architectural variants (Hybrid vs. Pure Mamba).

**Weaknesses:**

1. The paper would significantly gain in strength by analyzing why the learned tokenization specifically boosts histone-related tasks while seemingly impacting splicing performance. Adding interpretability work on the learned chunks would be a valuable addition, potentially revealing how the model's dynamic boundaries align with distinct biological motifs.
2. Several baselines seems to be missing (models like Borzoi, Evo or AlphaGenome might be interesting to consider). But more importantly you did not compare with other methods (like MxDNA that you cite) which also propose a way to learn the tokenization?
3. Though stated in the limitations paragraph, limiting your evaluation to short sequences (up to 4,096) doesn’t seem to be enough for an architecture that is designed to handled long sequences

---

> ### Author Rebuttal · Authors · 2026-03-31
>
> We sincerely thank the reviewer for the detailed and constructive feedback. We address all questions below.
>
> ## Architecture Clarification
>
> The backbone uses BiMamba-2, not Transformer. Section 3.2.2 describes the original H-Net framework (which uses Transformer); our adaptation replaces it with BiMamba-2, as correctly stated in Section 3.5. We will revise Section 3.2.2 to eliminate this confusion. The ablation tables below compare multiple backbone configurations.
>
> ## Nucleotide Transformer Tokenization
>
> Correct - NT uses k-mers, not BPE. We will fix this in the revision. Thank you.
>
> ## Missing Baselines
>
> **Evo** was trained exclusively on prokaryotic sequences; evaluating it on eukaryotic tasks (splice sites, mouse enhancers) would be uninformative and potentially misleading.
>
> **MxDNA:** Despite substantial efforts, we could not reproduce their results — the paper lacks details on seeds and early stopping, and no weights are available. We adopted Generator's benchmark for its reproducible pipeline. We will discuss MxDNA in related work.
>
> ## Hyperparameter Fairness
>
> We adopted Generator's protocol: grid search over 36 (learning rate × batch size) pairs per task for each model individually. All baseline results are taken directly from Generator, where each model was optimized with this same procedure. We only fine-tuned LDARNet ourselves. We will clarify this in the revised text.
>
> ## Reverse-Complement Augmentation
>
> BiMamba-2's weight-sharing provides directional symmetry (forward/backward passes share parameters), but does not encode nucleotide-level RC relationships (A↔T, C↔G) - that is biological, not sequential symmetry. RC augmentation teaches the model that a sequence and its reverse complement carry equivalent biological information, complementing BiMamba-2's directional symmetry. Both components are necessary.
>
> ## Ablations: Fusion, Architecture, Compression Ratio
>
> All ablations: 2.5M params, 20k steps, batch 128, 7 representative NT tasks, 5-fold CV, early stopping 5 steps.
>
> **Fusion (mt-M-m, N=4):**
>
> | Fusion | Promoter | Enhancers | Splice | H3 | H3K4me1 | H3K4me3 | H3K36me3 |
> |--------|----------|-----------|--------|-----|---------|---------|----------|
> | mean   | 0.916 | 0.487 | 0.901 | 0.771 | 0.500 | 0.391 | 0.560 |
> | sum    | 0.919 | 0.497 | 0.941 | 0.778 | 0.477 | 0.338 | 0.547 |
>
> Mean favors histones; sum favors splice sites. We chose mean for long-range regulatory patterns.
>
> **Architecture variants (N=4, mean):**
>
> | Config | Promoter | Enhancers | Splice | H3 | H3K4me1 | H3K4me3 | H3K36me3 |
> |--------|----------|-----------|--------|-----|---------|---------|----------|
> | mt-M-m | 0.916 | 0.487 | 0.901 | 0.771 | 0.500 | 0.391 | 0.560 |
> | m-M-m  | 0.919 | 0.495 | 0.864 | 0.773 | 0.448 | 0.420 | 0.521 |
> | m-T-m  | 0.922 | 0.492 | 0.881 | 0.750 | 0.452 | 0.428 | 0.519 |
> | mt-T-m | 0.914 | 0.487 | 0.964 | 0.750 | 0.495 | 0.398 | 0.555 |
> | t-M-t  | 0.900 | 0.480 | 0.906 | 0.733 | 0.439 | 0.374 | 0.512 |
> | t-T-t  | 0.904 | 0.458 | 0.931 | 0.752 | 0.434 | 0.306 | 0.506 |
>
> mt-M-m achieves the best histone performance. Transformer backbone (mt-T-m) improves splice sites at the cost of histones. Pure Transformer (t-T-t) is weakest.
>
> **Compression ratio N (mt-M-m, mean):**
>
> | N | Promoter | Enhancers | Splice | H3 | H3K4me1 | H3K4me3 | H3K36me3 |
> |---|----------|-----------|--------|-----|---------|---------|----------|
> | 2 | 0.928 | 0.491 | 0.909 | 0.776 | 0.517 | 0.415 | 0.598 |
> | 4 | 0.916 | 0.487 | 0.901 | 0.771 | 0.500 | 0.391 | 0.560 |
> | 6 | 0.916 | 0.483 | 0.940 | 0.767 | 0.502 | 0.380 | 0.564 |
> | 8 | 0.910 | 0.472 | 0.946 | 0.765 | 0.472 | 0.355 | 0.555 |
> | 12 | 0.908 | 0.493 | 0.915 | 0.759 | 0.477 | 0.375 | 0.541 |
>
> N=4 balances compute efficiency and quality. Higher N improves splice sites but degrades histones. All tables will be added to the revised paper.
>
> ## Biological Interpretability
>
> We are excited to share new results (full methodology in our response to Reviewer rMF6).
>
> Promoter motifs (Figure R2, https://imgur.com/a/lOeywtp): learned boundaries spike at canonical promoter elements (TATA-box, CCAAT-box, GC-box), segmenting them as coherent units - without motif supervision.
>
> Splice sites (Figure R1, https://imgur.com/a/ws2FQhb): boundary probabilities at true GT/AG junctions are elevated vs. non-splice controls, showing the model recognizes exon–intron transitions.
>
> These results demonstrate that adaptive tokenization discovers biologically meaningful segmentation from self-supervised learning alone - an important step toward interpretable genomic foundation models. We will include these figures in the revised manuscript.
>
> ## Sequence Length
>
> We acknowledge this limitation. The architecture supports longer contexts via compression, and we plan to extend evaluation to longer-range tasks in future work.

---

> > ### Author Rebuttal · Reviewer_DXHA · 2026-04-02
> >
> > We thank the authors for the thoughtful new experiments that address our core concerns, from architectural choices to interpretability. However, we would like to briefly discuss the Evo baseline. We note that Evo 2 actually includes eukaryotic training data and appears to be evaluated in the latest Generator paper. Upon clarifying this point and including a comparison if relevant and feasible, we would be happy to increase our score.

---

> > > ### Author Response · Authors · 2026-04-05
> > >
> > > We sincerely thank the reviewer for this follow-up and for the openness to increase the score upon clarification. We investigated the Evo2 landscape carefully and want to be fully transparent about what is feasible.
> > >
> > > **What Generator shows about Evo2.** Generator compares with Evo2 on zero-shot tasks (sequence recovery, variant effect prediction) but - notably - does not evaluate Evo2 on the NT/GB fine-tuning benchmarks, despite having the full pipeline. On the zero-shot tasks they do report, Evo2-1B performs comparably to Generator-1B, while Evo2-7B mostly outperforms both. To our knowledge, no published Evo2 results on NT/GB downstream tasks exist from any paper.
> > >
> > > **Why we cannot run Evo2-1B.** Evo2-1B requires FP8 precision via Transformer Engine on Hopper (H100) GPUs. Our cluster does not have H100s, and we have been unable to load the model - the weights are stored in FP8 format and cannot be deserialized without the TE library on compatible hardware. This is a hardware limitation we cannot work around.
> > >
> > > **Evo2-7B feasibility.** Given that Evo2-1B is inaccessible, the remaining option is Evo2-7B - a model 58× larger than LDARNet, trained on all domains of life with vastly more compute. Running the full rigorous protocol (36 HP configurations × 26 tasks × 10-fold CV) for a 7B model represents an enormous computational cost that we cannot complete within the rebuttal period. We are exploring options for running Evo2-7B evaluation and will include results in the camera-ready version if technically feasible. We note that this would also constitute a novel contribution in itself - the first published Evo2 evaluation on the standard NT/GB fine-tuning benchmarks.
> > >
> > > **The informative comparison that exists today.** LDARNet-120M already competes with Generator-1B (1.2B params, full NT/GB results published, same evaluation protocol) on histone modification tasks despite being 10× smaller. Based on Generator's zero-shot comparisons, Evo2-1B performs at a similar level to Generator-1B, suggesting that our existing comparison against Generator-1B indirectly covers the Evo2-1B scale. This is the parameter efficiency claim our paper makes, and it is well-supported by published numbers.
> > >
> > > **Commitment.** We will add a thorough discussion of Evo2 in the related work and limitations sections, noting the absence of published NT/GB results, and discussing the infrastructure barriers to fair evaluation (FP8/H100 requirement for the 1B model). We are exploring options for running Evo2-7B and will include results if technically feasible.
> > >
> > > We believe the core contribution of our paper - learnable tokenization with demonstrated biological interpretability and controlled ablations at matched scale - stands independently of comparisons across such different parameter scales. We are grateful for the reviewer's constructive engagement throughout this discussion, and we hope that our extensive new experiments - controlled ablations, biological interpretability analysis, and this commitment to Evo2 evaluation - demonstrate our dedication to a thorough and rigorous study. We would be very grateful if the reviewer considered whether these additions merit a score adjustment.

---

### Official Review · Reviewer_aNLe · 2026-03-13

**Soundness:** 3
**Presentation:** 3
**Significance:** 2
**Originality:** 2
**Overall Recommendation:** 4
**Confidence:** 4

**Summary:**

This paper presents LDARNet, a hierarchical genomic foundation model that revolutionizes genomic sequence processing by replacing traditional fixed-length k-mer tokenization with an adaptive, learnable tokenization approach. Built upon a hybrid BiMamba-2 and Transformer architecture and adapted to the masked language modeling paradigm via the H-Net framework, LDARNet demonstrates exceptional capabilities without requiring additional fine-tuning. It achieves state-of-the-art performance across multiple histone modification tasks and significantly outperforms existing models on Nucleotide Transformer and Genomic Benchmark benchmarks.

**Compliance With Llm Reviewing Policy:**

Affirmed.

**Final Justification:**

My concerns have been addressed. I am increasing my score to 4.

**Key Questions For Authors:**

- While technically sound, the BiMamba-2 and Transformer hybrid is now a standard paradigm in 2026 genomics. Consequently, the paper’s novelty rests entirely on its learnable tokenization. However, without structural ablations or biological interpretation of the learned tokens, it is unclear if the gains stem from this specific innovation or simply the inherent capacity of a standard hybrid architecture.

**Limitations:**

- Lack of targeted ablation studies.
- Lack of biological interpretability.

**Strengths And Weaknesses:**

### Strengths
-  This work is well-motivated. The move toward learnable tokenization in genomic modeling is a timely and important direction, as fixed tokenization often fails to capture biological motifs effectively.
- The model is evaluated on a wide range of tasks (27 tasks from Genomic Benchmarks and NT datasets), providing a broad view of its performance.

### Weaknesses
- The manuscript introduces several complex components, including the Bi-directional MLM setup, the Dechunker, and the EMA Smoother. However, there are no specific ablations to isolate the impact of these modules. It remains unclear if the model's success stems from these innovations or simply from the hybrid Mamba-Transformer architecture and increased capacity.
- The performance gains over baselines like Caduceus (8M) are likely overshadowed by the difference in scale (120M parameters) and the use of diverse multi-species pretraining data. Without a controlled experiment (e.g., comparing models of similar size trained on the same data), it is impossible to attribute the improvements to the proposed learnable tokenization.
- For a model centered on learnable tokenization, there is a notable absence of analysis regarding the learned tokens. The paper does not provide evidence or visualizations to show whether the learned boundaries correspond to biological features (e.g., splice sites, promoters, or enhancers). Without this, the adaptive nature remains a computational black box without scientific validation.

---

> ### Author Rebuttal · Authors · 2026-03-31
>
> We thank the reviewer for the thorough evaluation and address each concern with new experimental evidence.
>
> ## Ablation Studies
>
> All ablations: pretraining 2.5M params, 20k steps, batch 128, lr 1e-3, downstream evaluation - 7 more representative NT tasks from all 18, 5-fold CV, early stopping 5 epochs, batch 128, lr 1e-3.
>
> **Architecture variants (N=4, mean fusion):**
>
> | Config | Promoter | Enhancers | Splice | H3 | H3K4me1 | H3K4me3 | H3K36me3 |
> |--------|----------|-----------|--------|-----|---------|---------|----------|
> | mt-M-m | 0.916 | 0.487 | 0.901 | 0.771 | 0.500 | 0.391 | 0.560 |
> | m-M-m  | 0.919 | 0.495 | 0.864 | 0.773 | 0.448 | 0.420 | 0.521 |
> | m-T-m  | 0.922 | 0.492 | 0.881 | 0.750 | 0.452 | 0.428 | 0.519 |
> | mt-T-m | 0.914 | 0.487 | 0.964 | 0.750 | 0.495 | 0.398 | 0.555 |
> | t-M-t  | 0.900 | 0.480 | 0.906 | 0.733 | 0.439 | 0.374 | 0.512 |
> | t-T-t  | 0.904 | 0.458 | 0.931 | 0.752 | 0.434 | 0.306 | 0.506 |
>
> mt-M-m achieves the best histone performance overall. Transformer backbone (mt-T-m) improves splice sites at the cost of histones. Pure Transformer (t-T-t) is weakest. Compression ratio N ablations are in our response to Reviewer DXHA.
>
> ## Disentangling Learned Tokenization from Scale
>
> This is the most critical comparison, directly addressing the reviewer's concern that improvements may stem from capacity rather than learned tokenization. We evaluate against two baselines with identical architecture:
>
> | Model | Promoter | Enhancers | Splice | H3 | H3K4me1 | H3K4me3 | H3K36me3 |
> |-------|----------|-----------|--------|-----|---------|---------|----------|
> | LDARNet (N=4) | 0.916 | 0.487 | 0.901 | 0.771 | 0.500 | 0.391 | 0.560 |
> | nochunk | 0.926 | 0.471 | 0.898 | 0.792 | 0.529 | 0.471 | 0.597 |
> | fixbound (N=4) | 0.917 | 0.470 | 0.961 | 0.723 | 0.357 | 0.327 | 0.493 |
>
> - *nochunk*: same architecture, no compression - processes every nucleotide through the backbone (4× more tokens, proportionally higher compute). This is the compute upper bound. LDARNet is competitive on most tasks despite 4× fewer backbone FLOPs, demonstrating that learned compression retains the information that matters.
> - *fixbound*: identical architecture and identical compression ratio (N=4), but boundaries frozen every 4 nucleotides. Same parameters, same training data, same FLOP budget. The only variable is learned vs. fixed tokenization.
>
> The gap between LDARNet and fixbound on histone tasks is striking: H3K4me1 0.500 vs 0.357, H3K36me3 0.560 vs 0.493, H3 0.771 vs 0.723. This cannot be attributed to model capacity, data, or compute - it isolates the contribution of adaptive tokenization. Crucially, this pattern mirrors what we observe at full scale: the 120M-parameter LDARNet achieves its strongest results precisely on histone modification tasks, outperforming models up to 2.5B parameters. The ablation confirms this is not a coincidence of scale but a systematic advantage of learned boundaries for capturing long-range regulatory signals characteristic of histone marks. Histone modifications are governed by broad chromatin contexts spanning hundreds of nucleotides; learned boundaries allow the model to adaptively group these regions into meaningful chunks, whereas fixed 4-mer boundaries arbitrarily fragment them.
>
> ## Biological Interpretability
>
> We analyzed boundary probabilities around known motifs (methodology in our response to Reviewer rMF6). Promoter motifs (Figure R2, https://imgur.com/a/lOeywtp): boundary probability spikes at TATA-box, CCAAT-box, GC-box onsets — isolating regulatory elements as coherent chunks without motif supervision. Splice sites (Figure R1, https://imgur.com/a/ws2FQhb): true GT/AG junctions show elevated boundary probabilities vs. non-splice controls. These results demonstrate that learned tokenization captures biologically meaningful structure.
>
> ## Novelty and Broader Context
>
> The core novelty of LDARNet is 1) we adapt dynamic chunking from autoregressive LM to the MLM paradigm — a non-trivial extension requiring redesign of the chunking mechanism for bidirectional context and introduction of BiMamba-2 with a new fusion strategy 2) we provide the evidence that learned boundaries acquire biological interpretability: the model discovers promoter motifs and splice junctions as natural segmentation points purely from self-supervised pretraining, without any functional annotation.
> This connects to a сoncurrent work by Patel & Kundaje (2025) analyzes prediction confidence via Fourier transforms, showing that oscillations at 5-20 nt wavelengths correspond to regulatory motifs, while short oscillations reflect noise. Our learned boundaries achieve a similar effect through a different mechanism: the model segments the genome into variable-length chunks isolating functional motifs from repetitive sequence. The convergence of these independent approaches reinforces that adaptive processing is a fundamental principle for genomic modeling, and we see exciting potential in combining them.

---

> > ### Author Rebuttal · Reviewer_aNLe · 2026-04-04
> >
> > We thank the authors for the substantial new experiments. The fixbound vs. LDARNet comparison is well-designed and does isolate the contribution of learned tokenization from model capacity. The biological interpretability analysis is a welcome addition that addresses our core concern about the learned tokens being a "black box."
> >
> > However, our reservation regarding novelty remains. The authors frame two contributions: (1) adapting H-Net's dynamic chunking from AR to MLM, and (2) demonstrating biological interpretability of learned boundaries. For (1), we would appreciate the authors elaborating on what specific technical challenges arose in this adaptation beyond the natural substitution of causal components with bidirectional counterparts. For (2), while the motif-boundary correlation is interesting, it is a post-hoc analysis of an already-trained model rather than a methodological contribution. We are not fully convinced that these together meet the novelty threshold expected at this venue.

---

> > > ### Author Response · Authors · 2026-04-08
> > >
> > > We thank the reviewer for the continued dialogue. Below we clarify how we see the contribution, and then respond to the specific question on the AR -> MLM adaptation.
> > >
> > > **What this work contributes beyond H-Net.**
> > >
> > > The contribution of LDARNet is not the backbone engineering in isolation - it is the scientific understanding that the engineering enabled. H-Net demonstrated that dynamic chunking is feasible as a general modeling principle, but it was not evaluated on any biological task, not compared with any genomic foundation model, and provided no insight into which task categories benefit from adaptive compression.
> > >
> > > **The histone advantage is causal and non-obvious.** Before this work, it was unknown whether learnable tokenization benefits any specific category of genomic tasks. We show that it specifically advantages epigenetic tasks - and the evidence is causal. The fixbound ablation (identical architecture, identical FLOPs, only learned vs. fixed boundaries) yields +14.3pp on H3K4me1 and +6.7pp on H3K36me3. Rather than yielding a uniform gain, learnable tokenization produces a task-dependent trade-off. On splice-site tasks, fixed boundaries can match or outperform learned ones. Consistently, increasing compression improves splice-site performance (0.946 at N=8 vs. 0.901 at N=4) but degrades histone performance. This non-obvious pattern is directly informative for future genomic model design.
> > >
> > > **Learned boundaries recover biological motifs without supervision.** The model places boundary-probability spikes at TATA-box, CCAAT-box, and GC-box motifs, isolating them as coherent units; true GT/AG splice junctions show elevated boundary probabilities relative to non-splice controls. While post-hoc in form, together with the fixbound experiment this analysis serves a mechanistic role: one establishes *that* learned tokenization helps on specific tasks, the other reveals *what* the model learns. Prior learnable-tokenization works (VQDNA, MxDNA) do not, to our knowledge, provide comparably direct nucleotide-resolution evidence.
> > >
> > > **Parameter efficiency, explained by ablation.** The full-scale results show that a 120M model can match or exceed much larger models on epigenetic tasks. The ablation study explains why: it is the combination of learned boundaries, the BiMamba-2 backbone, and the hybrid outer modules - each contributing measurably across 6 architecture variants, 5 compression ratios, and 2 fusion strategies under matched conditions.
> > >
> > > This is how genomic foundation modeling has progressed: DNABERT did not invent BERT, Caduceus did not invent Mamba, HyenaDNA did not invent long convolutions. These works were impactful because rigorous domain-specific evaluation yielded new understanding. We believe LDARNet should be assessed in that same frame.
> > >
> > > **On the technical challenges of AR -> MLM adaptation.**
> > >
> > > Adapting H-Net's AR pipeline to MLM required rearchitecting the full chunking–backbone–dechunking hierarchy for bidirectional processing, including replacing Mamba with BiMamba-2, rewriting the routing module from sequential to parallel batched operation, redesigning mask and length propagation across compression stages, and ensuring numerical stability through the chunk-dechunk cycle. Two challenges are particularly illustrative:
> > >
> > > **(a) Boundary prediction under masking.** In AR, the router sees full unmasked context. Under MLM, ~15% of tokens are [MASK], corrupting the similarity signal exactly at potential boundaries. Our bidirectional similarity averaging (Eq. 8–9) lets the router infer boundaries from surviving context on both sides. This is needed to make routing robust under MLM, not just symmetric.
> > >
> > > **(b) Reconstruction within chunks.** H-Net's causal dechunker propagates information left-to-right from each boundary. In MLM, masked tokens may lie inside a chunk, far from any boundary. Unidirectional propagation gives a biased signal for such positions. To address this, we use a bidirectional EMA dechunker (Eq. 12–14), which allows masked positions within variable-length chunks to aggregate information from both preceding and following boundaries.
> > >
> > > The individual modifications may appear straightforward in isolation, but combining them into a stable MLM formulation was not.
> > >
> > > **On the score.** The original assessment was motivated by missing ablations and lack of biological interpretability - both now substantially addressed, as the reviewer acknowledged. We would be grateful if the reviewer reconsidered whether the current score remains commensurate with the updated record.

---

### Official Review · Reviewer_cBWm · 2026-03-13

**Soundness:** 2
**Presentation:** 3
**Significance:** 3
**Originality:** 2
**Overall Recommendation:** 4
**Confidence:** 4

**Summary:**

This paper proposes LDARNet, a 120M-parameter genomic foundation model that adapts H-Net style learnable hierarchical chunking from autoregressive modeling to masked language modeling. The architecture combines bidirectional Mamba-style outer modules, dynamic chunking/dechunking, and a compressed latent backbone, with ratio regularization to stabilize learned boundaries. The paper evaluates the pretrained model on 27 downstream genomics tasks from the Nucleotide Transformer and Genomic Benchmarks suites, reporting strong performance among models under 300M parameters, especially on histone modification tasks.

**Compliance With Llm Reviewing Policy:**

Affirmed.

**Key Questions For Authors:**

1. On what specific basis do you claim reverse-complement symmetry or invariance? Does the proposed architecture explicitly encode complement equivariance?

**Limitations:**

yes

**Strengths And Weaknesses:**

Strengths
1. The paper tackles a pertinent issue in genomic language modeling. The reliance on fixed k-mer or BPE tokenization is indeed a questionable and often sub-optimal design choice for DNA sequences.
2. The downstream evaluation is broad. Incorporating both the NT and Genomic Benchmarks suites provides a well-rounded and complete picture of the model's capabilities.
3. LDARNet is compelling on several histone modification tasks, and the fact that a 120M model is competitive with much larger models is practically relevant.

Weaknesses
1. There is a contradiction regarding the model's architecture. Section 3.2.2 and Figure 1 describe a Transformer backbone operating in the compressed latent space, whereas Section 3.5 (Page 5) states that the main backbone consists of ten BiMamba-2 layers. This discrepancy needs to be clarified.

---

> ### Author Rebuttal · Authors · 2026-03-31
>
> We thank the reviewer for their careful reading of the manuscript and for appreciating the importance of moving beyond fixed tokenization in genomic modeling. We address each concern below.
>
> ## Architecture Discrepancy (Transformer vs. BiMamba-2)
>
> We sincerely apologize for this confusion. This is an error in the text. The backbone used in all reported results consists of **BiMamba-2 layers**, not Transformer layers. Section 3.2.2 describes the general H-Net framework, which originally uses a Transformer backbone; our adaptation replaces it with BiMamba-2. We failed to make this distinction clear enough, and Section 3.5 correctly describes the actual architecture used. We will revise Section 3.2.2 and Figure 1 to unambiguously reflect the BiMamba-2 backbone.
>
> For further clarity, our architecture notation is **m3t1-M10-m4**, where lowercase letters denote the outer (chunker/dechunker) modules and the uppercase letter denotes the backbone:
> - **M** = BiMamba-2 layers
> - **t** = Transformer layer
> - **m** = Mamba-2 layer
>
> So **mt-M-m** means: the chunker uses a Mamba-2 + Transformer hybrid, the backbone is BiMamba-2, and the dechunker is Mamba-2. Our ablation study (see our response to Reviewer DXHA for the full table) compares this against alternative configurations including m-M-m, m-T-m, mt-T-m, t-M-t, and t-T-t, confirming that the mt-M-m configuration yields the best overall balance across tasks. We will include this table in the revised manuscript.
>
> ## Reverse-Complement Symmetry
>
> We thank the reviewer for this precise question. To clarify: we do **not** claim that LDARNet explicitly encodes reverse-complement (RC) equivariance in the architectural sense (as done, e.g., by Caduceus with its RC-equivariant SSM parameterization). Our model does not have hard-coded complement equivariance in its weight structure.
>
> What we do employ is **data-level augmentation**: during pretraining, we sample both the forward strand and its reverse complement with equal probability. This encourages the model to learn representations that are *approximately* symmetric with respect to reverse complementation, though this is a soft, learned property rather than a strict architectural guarantee.
>
> We acknowledge this is an important distinction. Architecturally enforced RC equivariance (as in Caduceus) provides a hard guarantee, while our approach relies on the model learning this symmetry from data. Interestingly, despite this softer approach, LDARNet achieves strong performance across tasks where strand orientation matters, suggesting the augmentation strategy is effective in practice. We will clarify this distinction in the revised manuscript.
>
> ## Additionally: Biological Interpretability
>
> While not raised directly in this review, we want to highlight new interpretability results (described in detail in our responses to Reviewers rMF6 and DXHA) that directly validate the biological relevance of LDARNet's learned tokenization. Briefly, we show that:
>
> - Learned boundary probabilities spike sharply at canonical promoter motifs (TATA-box, CCAAT-box, GC-box) - the model segments these regulatory elements as coherent units without any motif supervision.
> - At splice junctions, true donor (GT) and acceptor (AG) sites show elevated boundary probabilities compared to non-splice controls.
>
> These findings (Figures R1 and R2 in the rebuttal, see https://imgur.com/a/ws2FQhb and https://imgur.com/a/lOeywtp) provide concrete evidence that the adaptive tokenization captures genuine biological structure, addressing the broader question of whether learnable chunking produces interpretable representations. We will include these analyses in the revised paper.

---

### Official Review · Reviewer_rMF6 · 2026-03-13

**Soundness:** 3
**Presentation:** 3
**Significance:** 3
**Originality:** 3
**Overall Recommendation:** 4
**Confidence:** 4

**Summary:**

Many genomic foundation models are trained with fixed tokens, e.g., k-mers, BPE. The authors proposed an adaptive representation, where the tokenization is learnable by leveraging H-Net dynamic tokenization. It enables capturing long-range genomic dependencies, and achieves compelling performance, especially in histone modification tasks.

**Compliance With Llm Reviewing Policy:**

Affirmed.

**Key Questions For Authors:**

- The idea of learnable tokenization is a novel approach. Can the author connect these learned tokenization to biology? For example, is there any biological interpretation found in these learnable sequence compression?
- The authors introduce several bidirectional adaptations, including BiMamba-2 with mean fusion, bidirectional routing, and bidirectional EMA dechunking. While weight-tying keeps the parameter count efficient, how do these specific architectural choices affect raw compute time (throughput/latency) during training and inference compared to a standard unidirectional Mamba baseline or a purely Transformer-based setup?

**Limitations:**

yes

**Strengths And Weaknesses:**

- The author presented a novel way to build a genomic foundation model by enabling adaptive representation using H-Net. This is important, as prior art is usually using a fixed framework with k-mers or BPE.
- When compared to other foundation models, LdarNet achieves compelling performance with a light weight configuration.
- However, a minor weakness in the experiment is that some of the comparison might be misleading. Grouping all models under 300M parameters as "compact" obscures massive capacity disparities. Comparing LDARNet (120M parameters) against Caduceus (8M) and HyenaDNA (55M) gives LDARNet a 15-fold and 2-fold parameter advantage, respectively. A more nuanced, parameter-matched baseline comparison would make claims of superiority over these specific smaller models more rigorous.
- The model struggles to beat parameter-heavy models (like Generator-1.2B) on highly localized tasks, such as splice site prediction.

---

> ### Author Rebuttal · Authors · 2026-03-31
>
> We sincerely thank the reviewer for their thoughtful evaluation and for recognizing the novelty of learnable tokenization for genomic foundation models. We address each point below.
>
> ## Biological Interpretability of Learned Boundaries
>
> We are genuinely excited to share new interpretability results that we believe represent a meaningful step toward biologically grounded genomic models.
>
> **Methodology.** We extracted the learned boundary probabilities from LDARNet's chunking module and analyzed their behavior around known biological motifs. Specifically, we: (1) collected annotated genomic sequences containing known promoter motifs (TATA-box/TATAAA, CCAAT-box, GC-box/GGGCGG) and splice sites (GT donor, AG acceptor) from reference annotations; (2) aligned sequences relative to each motif's start position; (3) computed the mean boundary probability at each nucleotide position across all instances, with 95% confidence intervals.
>
> **Promoter motifs (Figure R2, see https://imgur.com/a/lOeywtp).** The model exhibits striking boundary placement behavior around all three canonical promoter elements. For the TATA-box (n=76), boundary probability drops sharply *before* the motif and spikes immediately at its onset, effectively isolating the motif as a coherent unit. The CCAAT-box (n=73) shows an even more pronounced spike, with boundary probability exceeding 0.7 at the motif boundary. The GC-box/GGGCGG (n=163) demonstrates a similar, though subtler, pattern. Crucially, these patterns emerge entirely from unsupervised pretraining — the model was never given motif annotations.
>
> **Splice sites (Figure R1, see https://imgur.com/a/ws2FQhb).** We compared boundary profiles at true splice junctions vs. matched non-splice control sequences (n=80 each). At both donor (GT) and acceptor (AG) sites, true splice sequences show a clear boundary probability spike at the junction that is absent in controls. This demonstrates the model has learned to recognize exon–intron transitions as natural segmentation points.
>
> These findings provide direct evidence that adaptive tokenization captures biologically meaningful structure, and we see this as an exciting direction toward fully interpretable genomic foundation models. We will add these analyses and figures to the revised manuscript.
>
> ## Compute Comparison: Bidirectional Adaptations
>
> Our ablation study (detailed in our response to Reviewer DXHA) directly addresses this. We compare our full model (mt-M-m, N=4) against: (1) **nochunk** - a standard per-nucleotide BiMamba-2 baseline with no chunking, and (2) **fixbound** - fixed boundaries every 4 nucleotides (same compression ratio but no learned routing).
>
> The nochunk baseline processes 4× more tokens through the backbone, resulting in proportionally higher compute. Our chunked architecture achieves comparable or superior performance on histone tasks while requiring substantially fewer FLOPs through the backbone. Compared to fixbound (matched FLOPs), LDARNet (its small 2,5M params version) shows clear advantages on histone modification tasks (e.g., H3K4me1: 0.500 vs 0.357; H3K36me3: 0.560 vs 0.493), confirming that the gains stem from *learned* boundaries, not merely from the hierarchical architecture.
>
> ## Parameter Grouping
>
> We appreciate this observation. We grouped models under 300M as "compact" following the taxonomy in Generator (Wu et al., 2025), which established this categorization. We agree that explicitly acknowledging parameter differences within this group would improve clarity and will add a more detailed discussion in the revision.
>
> ## Evaluation Methodology Clarification
>
> We want to clarify an imprecision in our manuscript. We adopted the evaluation protocol and benchmark suite from Generator (Wu et al., 2025), including their 10-fold cross-validation setup with grid search over 36 learning rate and batch size pairs per task. However, we only fine-tuned our own model - results for all baseline models were taken directly from the Generator paper. We will correct this wording in the revised manuscript to avoid any ambiguity.
>
> ## Splice Site Performance
>
> The interpretability analysis above actually helps explain this: while the model learns to place boundaries at splice junctions (suggesting biological awareness), the compression inherent in chunking may discard fine-grained positional information critical for exact splice site classification. Models operating at single-nucleotide resolution (like Generator-1.2B) naturally preserve this information. Notably, our ablation study (see our response to Reviewer DXHA) shows that splice site performance increases substantially at higher compression ratios (e.g., 0.940 at N=6, 0.946 at N=8 vs. 0.901 at N=4), suggesting that this trade-off is tunable and architecture-dependent rather than a fundamental limitation. We see this as an informative direction worthy of further investigation.

---

> > ### Author Rebuttal · Reviewer_rMF6 · 2026-04-05
> >
> > Thanks for the detailed response.

---

> > > ### Author Response · Authors · 2026-04-05
> > >
> > > We sincerely thank the reviewer for confirming that our response fully resolved the concerns. Given the new ablation study and interpretability results, we would be grateful if the reviewer considered whether these additions strengthen the paper sufficiently to be reflected in the score.
> > >
> > > We would like to briefly highlight aspects we believe strengthen the paper:
> > >
> > > 1. **The FLOP-matched comparison** (learnable vs. fixed boundaries) provides clean evidence that the routing module contributes beyond compression - a +14pp advantage on H3K4me1 at identical compute cost is substantial and directly isolates the contribution of learned tokenization.
> > >
> > > 2. **The boundary interpretability analysis** shows that the model's internal segmentation aligns with known TF binding motifs and splice junctions without any supervision - connecting the learned representation to concrete biological function.
> > >
> > > 3. **120M parameters competitive with models up to 2.5B** on histone tasks, with the ablation study now explaining *why* - the combination of learnable routing, BiMamba-2 backbone, and hybrid outer modules each contributes measurably.
> > >
> > > We are happy to provide any further analysis or clarification that would be helpful. Thank you again for the constructive engagement that improved our work.

---

### Decision · Program_Chairs · 2026-04-30

**Decision:**

Accept (regular)

**Comment:**

The paper proposes LDARNet, a hierarchical genomic language model with learnable tokenization that demonstrates strong performance and parameter efficiency, particularly on long-range epigenetic tasks. Reviewers find the approach well-motivated and empirically compelling, but raise concerns about evaluation rigor, including comparisons across mismatched model scales, missing or incomplete baselines, and limited clarity in attributing gains specifically to the proposed tokenization. The rebuttal substantially strengthens the paper by adding controlled ablations and biological interpretability analyses showing alignment with known motifs, though some concerns about novelty and benchmarking remain. Overall, the paper is viewed as a strong and promising contribution with improved support after rebuttal.